# Central Arctic Ocean paleoceanography from ~50 ka to present, on the basis of ostracode faunal assemblages from SWERUS 2014 expedition

Laura Gemery[1], Thomas M. Cronin[1], Robert K. Poirier[1&2], Christof Pearce[3,4], Natalia Barrientos[3], Matt O'Regan[3], Carina Johansson[3], Andrey Koshurnikov,[5,6] Martin Jakobsson[3]

[1] U.S. Geological Survey, Reston, Virginia
[2] Rensselaer Polytechnic Institute, Department of Earth & Environmental Sciences, Troy, New York
[3] Department of Geological Sciences and Bolin Centre for Climate Research, Stockholm University, Stockholm, 10691, Sweden
[4] Department of Geoscience, Aarhus University, Aarhus, 8000, Denmark
[5] Tomsk National Research Polytechnic University, Tomsk, Russia
[6] Moscow State University, Geophysics, Russian Federation

*Correspondence to*: Laura Gemery (lgemery@usgs.gov)

Keywords: Arctic Ocean, Quaternary, Sea-ice history, Sediment cores, Paleobiological proxies, Benthic ostracode assemblages

Abstract

Late Quaternary paleoceanographic changes at the Lomonosov Ridge, central Arctic Ocean, were reconstructed from multicore and gravity cores recovered during the 2014 SWERUS-C3 Expedition. Ostracode assemblages dated by accelerator mass spectrometry (AMS) indicate changing sea-ice conditions and warm Atlantic Water (AW) inflow to the Arctic Ocean from ~50 ka to present. Key taxa used as environmental indicators include *Acetabulastoma arcticum* (perennial sea ice), *Polycope* spp. (variable sea ice margins, high surface productivity), *Krithe hunti* (Arctic Ocean deep water), and *Rabilimis mirabilis* (nutrient, AW inflow). Results indicate periodic seasonally sea-ice free conditions during Marine Isotope Stage (MIS) 3 (~57-29 ka), rapid deglacial changes in water mass conditions (15-11 ka), seasonally sea-ice free conditions during the early Holocene (~10-7 ka) and perennial sea ice during the late Holocene. Comparisons with faunal records from other cores from the Mendeleev and Lomonosov Ridges suggest generally similar patterns, although sea-ice cover during the last glacial maximum may have been less extensive at the new Lomonosov Ridge core site (~85.15°N, 152°E) than farther north and towards Greenland. The new data provide evidence for abrupt, large-scale shifts in ostracode species depth and geographical distributions during rapid climatic transitions.

1. Introduction

Environmental conditions are changing rapidly in the Arctic Ocean today, but a longer time perspective is necessary to assess and contextualize these changes and their contributing factors.  These changing conditions include sea ice extent and thickness (Stroeve et al., 2012, 2014; Laxon et al., 2013), as well as ocean temperature, stratification, circulation, chemistry, and ecology  (Polyakov et al, 2017; Moore et al, 2015; Chierici and Fransson 2009; Rabe et al., 2011;

Grebmeier et al., 2006, 2012; Wassmann et al., 2011). Sea ice extent and
thickness, in particular, are challenging parameters to reconstruct because most
sea ice proxies lack temporal and geographical resolution (Stein et al., 2012). Sea
ice extent and thickness, however, are very important variables because they
influence albedo, near-surface salinity, light levels, surface-to-seafloor organic
carbon flux, and other variables that are important to ecosystems. In fact, sea ice
exerts a primary control on Arctic biological and geochemical cycles (Anderson et
al, 2011), and sea ice changes are in part responsible for fast-feedback climate
changes during the geologic past (Polyak et al., 2010).
Before the last few decades, instrumental oceanographic records were relatively
sparse, and sediment proxy records provided insight into past sea-ice conditions
and ocean circulation changes from all regions of the Arctic. These records are
especially important for examining sea-ice history during past climate changes
before the availability of instrumental records. The composition and abundance of
marine microfossils preserved in many Arctic sediments provide an important
constituent that helps address the growth and decay of ice sheets. For example,
several excursions in records of oxygen and carbon isotopes of planktic
foraminifers from Arctic sediment cores have been interpreted as releases of
freshwater from collapsing continental ice sheets during glaciations and glacial
terminations (Stein et al., 1994; Nørgaard-Pedersen et al., 1998)
This paper examines temporal changes in microfossil shells from ostracode
indicator species that shed light on biological productivity and sea-ice extent during
the last ~50 ka, including Marine Isotope Stages (MIS) 3, the Last Glacial
Maximum (LGM, ~21 ka), the deglacial interval and the Holocene. Ostracoda are
bivalved Crustacea that inhabit Arctic marine habitats and whose assemblages
(Cronin et al., 1994, 1995, Poirier et al., 2012) and shell chemistry (Cronin et al.,
2012) have been used extensively as proxies to reconstruct Arctic
paleooceanography and sea-ice history (Cronin et al., 2010). Most ostracode
species are benthic in habitat and their ecology reflects bottom water
environmental conditions. Benthic ecosystems rely on biological productivity in the
upper water column, and so benthic biomass production and community structure
also reflect sea-ice cover and surface-to-bottom ecosystem links (Grebmeier and
Barry, 1991; Grebmeier et al., 2006).
Two pelagic/epipelagic ostracode taxa are used in this paper to indicate water
mass conditions. Sediment cores for this study were collected during the 2014
SWERUS-C3 (Swedish – Russian – US Arctic Ocean Investigation of Climate-
Cryosphere-Carbon Interactions) Leg 2 expedition from previously unstudied
regions of the Siberian margin and the Lomonosov Ridge. The radiocarbon-dated
records presented here are from 85.15°N, 152°E on the Lomonosov Ridge in the
central Arctic Ocean, a site located at ~800 m near the transition between Atlantic
Water and Arctic Intermediate Water in the modern Arctic Ocean. During prior
glacial-interglacial cycles, the region was influenced to various degrees by the
strength and depth penetration of Atlantic Water. For example, during glacial
intervals when thick ice shelves covered much of the Arctic Ocean (Jakobsson et
al., 2016), Arctic Intermediate Water warmed (Cronin et al., 2012) and likely
entrained to greater water depths (Poirier et al., 2012). Consequently, the new
results, when compared to published faunal records from other regions of the
Arctic Ocean (Fig. 1a, Table 1), show some regional differences but an overall
remarkable consistency in central Arctic faunal abundance changes during the late
Quaternary.
2. Arctic oceanography
The Arctic Ocean is strongly stratified, with distinct water masses separated by
vertical changes in salinity and temperature (Figure 1b). The following summary of
Arctic water masses and circulation is taken from Aagaard and Carmack (1989),
Anderson et al. (1994), Jones (2001), Olsson and Anderson (1997), Rudels et al.
(2012 and 2013). Arctic Ocean water masses include a fresh, cold Polar Surface
Water layer ([PSW], T= ~0°C to -2°C, S= ~32 to 34), found between ~0 and 50 m.
The PSW is characterized by perennial ice in most regions and seasonal sea ice in
the margins of the Arctic Ocean. Beneath the sea-ice cover, a strong halocline
separates the PSW from the underlying warmer, denser water mass of North
Atlantic origin (Atlantic Water [AW], ~200 to 1000 m, T= >0°C, S= ~34.6 to 34.8).
One branch of the AW flows into the Arctic Ocean from the Nordic seas along the
eastern Fram Strait off the west coast of Spitsbergen and another branch flows
through the Barents Sea. An intermediate-depth water mass below the AW in the
Eurasian Basin at ~1000-1500 m is called the Arctic Intermediate water ([AIW], T=
-0.5 to 0°C, S= ~34.6 to 34.8). Below 2000 m, the deep Arctic basins are filled with
Arctic Ocean Deep Water ([AODW], T= -1.0°C to -0.6°C, S= 34.9, Somavilla et al.,
2013). Bathymetry is a dominant factor governing circulation patterns for AW and
AIW, and a sharp front over the Lomonosov Ridge near the SWERUS-C3 core site
studied here partially isolates these waters in the Eurasian Basin from the
Canadian Basin (Fig. 1b).
In addition to Arctic Ocean stratification, other factors influence sea-ice decay and
growth over geologic time (i.e. Polyak et al., 2010). A recent study by Stein et al.
(2017) notes the importance of large-scale atmospheric circulation patterns, such
as the North Atlantic Oscillation (NAO) and Arctic Oscillation (AO), and radiative
forcing (i.e. solar activity) on Holocene sea ice thickness, extent and duration. The
NAO and AO influence changes of the relative position and strength of the two
primary Arctic Ocean surface-current systems, the Beaufort Gyre in the Amerasian
Basin and the Transpolar Drift in the Eurasian Basin (Fig. 1a; Rigor et al., 2002;
Stroeve et al., 2014). Data resulting from the SWERUS expedition will help
improve understanding of the spatial patterns of sea-ice and intermediate depth
circulation, given the extreme variability in sea ice in this region recently evident
from satellite records (Serreze and Stroeve, 2015; Stroeve et al., 2014), the
importance of the Transpolar Drift in sea ice export through Fram Strait (Polyak et
al., 2010; Smedsrud et al., 2017) and new evidence for the influence of inflowing
Atlantic Water on sea ice and "atlantification" of the Eurasian Basin (Polyakov et al.
137   2017).

3. Materials and methods
3.1 Core material and sample processing
Cores for this study were obtained during the September 2014 SWERUS-C3 (Leg
2) expedition to the eastern Arctic Ocean aboard Swedish Icebreaker *Oden*. Figure
1 shows the location of multicore SWERUS-L2-32-MC4 (85.14°N, 151.57°E, 837
m) and nearby gravity core SWERUS-L2-32-GC2 (85.15°N, 151.66°E, 828 m) on
the Lomonosov Ridge. These cores are hereafter referred to as 32-MC and 32-
GC, respectively. Both cores were stored at 4°C and sampled at the Department of
Geological Sciences, Stockholm University. Processing of the samples involved
washing the sediment with water through a 63-µm mesh sieve. Core 32-MC was
processed in Stockholm while 32-GC was processed at the U.S. Geological
Survey (USGS) laboratory in Reston, Virginia. Sediment samples (1-cm thick, ~30
g prior to processing) were taken every centimeter in 32-MC along its 32 cm
length. Section 1 (117 cm) of 32-GC was sampled every 2-3 cm (2-cm thick, ~45-
60 g wet weight).
After processing and oven drying the samples, the residual >125 µm size fraction
was sprinkled on a picking tray and ostracodes were removed to a slide. One
exception for expediency is that specimens of the genus *Polycope* were counted
and not removed from the sediment. A total of ~300 specimens were studied from
each sample of 32-MC. More detailed counts of some samples in 32-MC were
done periodically, where all specimens were picked and/or counted to ensure that
300 specimens provided a representative assemblage. In 32-GC, all specimens
were picked and/or counted in each sample. Ostracodes were present throughout
the entire studied intervals of both 32-MC and down to 62 cm in 32-GC. Planktic
and benthic foraminifers were also present in abundance but not studied.
3.2 Chronology, reservoir corrections and sedimentation
Nine radiocarbon ($^{14}$C) ages were obtained from core 32-MC using accelerator
mass spectrometry (AMS) (Fig. 2, Table 2). Most dates were obtained on mollusks
(*Nuculidae* and *Arcidae* spp.), except a few samples where mollusks and benthic
foraminifera were combined. Two ages from 32-GC were obtained using a
combination of mollusks, foraminifera and ostracode shells. The final age models
representing the two cores combined are based on all the calibrated $^{14}$C ages
listed in Table 2. Generally, ages >40 ka should be considered with caution
because of large uncertainties in the radiocarbon calibration curve and high
sensitivity to even extremely small levels of contamination. Calibration into
calendar years was carried out using Oxcal4.2 (Bronk Ramsey, 2009) and the
Marine13 calibration curve (Reimer et al., 2013), using a local marine reservoir
correction, ΔR, of 300±100 years. Because ΔR values for the central Arctic Ocean
were not constant during the last 50 ka, it is difficult to date pre-Holocene
sediments independently (Pearce et al., 2017; Hanslik et al., 2012), and improved
age models may be available in the future.
Patterns in ostracode assemblages in both cores were used to correlate cores 32-
MC and 32-GC and produce a composite faunal record, which led to a 3-cm offset
for core 32-GC. After adding the 3-cm offset to sample depths of 32-GC, the 32-
MC core chronology was applied down to 31.5 cm core depth (dated at 39.6 ka).
The average sedimentation rate at the core site was ~1.5 cm/ka, which is typical of
central Arctic Ocean ridges (Backman et al., 2004; Polyak et al., 2009).

The lower section of 32-GC, from 31.5 cm to 61 cm, is beyond the limit of
radiocarbon dating. However, the litho-stratigraphy of the gravity core can be
readily correlated to other records from the central Lomonosov Ridge, where
multiple dating techniques constrain the approximate positions of MIS 4 and 5
boundaries (Jakobsson et al., 2001; O'Regan, 2011). A correlation between
SWERUS-C3 32-GC and AO96/12-1PC was previously presented in Jakobsson et
al. (2016). The correlation is supported by the occurrences in 32-GC of the
calcareous nannofossil *E. huxleyii* (Fig. 2). Based on this longer-term correlation,
sediments between 31 and 61 cm are less than 50 ka. This age estimate is
consistent with previous work on the Lomonosov Ridge, revealing a prominent
transition from coarse-grained, microfossil-poor sediments (diamict) into
bioturbated, finer-grained, microfossiliferous sediments that occurred during MIS 3
at approximately 50 ka (Spielhagen et al., 2004; Nørgaard-Pederson et al., 2007).
4. Results and Discussion
4.1 Ostracode taxonomy and ecology
The SWERUS 32 cores contained a total of 13,767 ostracode specimens in 32-MC
and a total of 5,330 specimens in the uppermost 5-62 cm of 32-GC (the top few
centimeters below the seafloor were not recovered in the gravity core). The bottom
54 cm of 32-GC (section 1 from 63-117 cm) was barren of calcareous material.
Twenty-eight ostracode species were identified in 32-MC and 21 species were
identified in 32-GC. Supplementary Tables S1 and S2 provide all species and
genus census data for 32-MC and 32-GC, respectively. Data will also be
accessible at NOAA's National Centers for Environmental Information (NCEI,
https://www.ncdc.noaa.gov/paleo-search/). The primary sources of taxonomy and
ecology were papers by Cronin et al. (1994, 1995, 2010), Gemery et al. (2015),
Joy and Clark (1977), Stepanova (2006), Stepanova et al. (2003, 2007, 2010),
Whatley et al. (1996, 1998), and Yasuhara et al. (2014).
Podocopid ostracodes were identified at the species level except the genera
*Cytheropteron* and myodocopid *Polycope.* Table 3 provides a list of species
included in the genus-level groups, which was sufficient to reconstruct
paleoenvironmental changes. There are several species of *Cytheropteron* in the
deep Arctic Ocean but they are not ideal indicator species given their widespread
modern distributions. There are at least eight species of *Polycope* in the Arctic
Ocean, but juvenile molts of *Polycope* species are difficult to distinguish from one
another.  Most specimens in 32-MC and 32-GC belonged to *P. inornata* Joy &
Clark, 1977 and *P. bireticulata* Joy & Clark, 1977. Nonetheless, most *Polycope*
species co-occur with one another, are opportunistic in their ecological strategy,
and dominate assemblages associated with high surface productivity and organic
matter flux to the bottom (Table 4; Karanovic and Brandāo, 2012, 2016).
The relative frequency (percent abundance) of individual dominant taxa is plotted
in Figure 3 and listed in Supplementary Table S3. Abundances were computed by
dividing the number of individual species found in each sample by the total number
of specimens found. For 32-MC, using the algorithm for a binomial probability
distribution provided by Raup (1991), ranges of uncertainty ("error bars") were
calculated at the 95% fractile for the relative frequency in each sample to the
relative frequency of each species and the total specimen count of each sample at
a given core depth (Supplementary Table S4). Faunal densities were high enough
to allow comparisons from sample to sample, and Supplementary Table S4 lists
the density of ostracode specimens per gram of dry sediment, which averaged
>125 shells per gram sediment. For this study of the SWERUS-C3 32 cores, the
focus was on an epipelagic species (*Acetabulastoma arcticum),* a pelagic genera
(*Polycope* spp.), three benthic species *(Krithe hunti, Pseudocythere caudata,*
*Rabilimis mirabilis)* and a benthic genus *(Cytheropteron* spp.)*.* Table 4 provides an
overview of pertinent aspects of these species' ecology that have
paleoceanographic application.
4.2 Temporal patterns in ostracode indicator species from SWERUS-C3 32-
MC/GC
The faunal patterns in cores from the SWERUS-C3 32-MC/GC sites confirm faunal
patterns occurring over much of the central Arctic Ocean during the last 50 ka,
including MIS 3-2 (~50 to 15 ka), the last deglacial interval (~15 to 11 ka), and the
Holocene (~11 ka to present). Similar patterns are seen in both the multicore and
gravity core. Relative frequencies of indicator taxa in cores 32-MC and 32-GC (Fig.
3) show four distinct assemblages, which are referred to as informal faunal zones
following prior workers (Cronin et al., 1995; Poirier et al., 2012). These zones are
as follows: (1) *Krithe* zone (primary abundance up to 80% during ~45-42 ka and a
secondary abundance of 5-10% during ~42-35 ka); (2) *Polycope* zone (with
abundance of 50 to 75% during ~40-12 ka, also containing a double peak in
abundance of *P. caudata*); (3) *Cytheropteron-Krithe* zone (12-7 ka); and (4)
*Acetabulastoma arcticum* zone (~7 ka-present). This paper briefly discusses the
paleoceanographic significance of each period in the following sections 4.3 - 4.5
based on the comparison cores presented in Figs 4 and 5. Figures 4 and 5
compare the new SWERUS-C3 results from 32-MC with published data from box
and multicores from the Lomonosov and Mendeleev Ridges, respectively, covering
a range of water depths from 700 m to 1990 m. Most records extend back to at
least 45 ka, and the age model for each core site is based on calibrated
radiocarbon ages from that site (i.e. Cronin et al., 2010, 2013; Poirier et al., 2012).
In addition, section 4.6 discusses a potential new indicator species, *R. mirabilis*,
which exhibits distinct faunal migrations that coincide with *Krithe* zones in 32-
MC/GC. *R. mirabilis* lives on today's continental shelf but is found in limited
intervals in sediment cores that are far outside its usual depth and geographic
range. *R. mirabilis* migrations are documented not only in 32-MC/GC but in cores
96-12-1PC, HLY0503-06JPC, P1-94-AR-PC10, P1-92-AR-PC40, LOMROG07-04
and P1-92-AR-PC30.
4.3 MIS 3-2 (~50-15 ka)
A strong peak in the abundance of *Krithe hunti* (Fig. 3) is seen in 32-GC sediments
estimated to be ~45-42 ka in age. A similar peak of lower but still significant
abundance also occurs in sediments dated between 42 and 35 ka, and this peak is
consistent with other cores on the Mendeleev Ridge and particularly on the
Lomonosov Ridge (Figs 4, 5). Prior studies of Arctic ostracodes have shown that
*Krithe* typically signifies cold well-ventilated deep water and perhaps low food
supply (Poirier et al., 2012 and references therein). *Krithe* is also a dominant
component (>30%) of assemblages in North Atlantic Deep Water (NADW) in the
subpolar North Atlantic Ocean. Its abundance varies during glacial-interglacial
cycles, reaching maxima during interglacial and interstadial periods (Alvarez
Zarikian et al., 2009). Peaks in the abundance of *Krithe* in the Arctic Ocean
probably signify faunal exchange between the North Atlantic Ocean and the
Greenland-Norwegian Seas through the Denmark Strait and Iceland Faroes Ridge
and the central Arctic through the Fram Strait. In other Arctic Ocean cores, the
ostracode genus *Henryhowella* is often associated with *Krithe* sp. in sediments
dated between ~50 to 29 ka (MIS 3), and its absence in the 32-MC/GC cores may
reflect the relatively shallow depth at the coring site. While *Henryhowella* was
absent in records from this site, *R. mirabilis* abruptly appears and spikes to an
abundance of 60 percent at 40 ka, which coincides with the *Krithe* zone.
*A. arcticum* is present in low abundance (~5%) in sediment dated at ~42 to 32 ka
in 32-MC/GC (Fig. 3), signifying intermittent perennial sea ice. A second increase
in abundance of *A. arcticum* corresponds to a (modeled, mean, 2-sigma)
radiocarbon date of 21.6 ka. This suggests the location of this core may not have
been covered by thick ice during the LGM as long as other areas.
A *Krithe* to *Polycope* shift occurred at ~35-30 ka. This "K-P shift" is a well-
documented, Arctic-wide transition (Cronin et al., 2014) that has
paleoceanographic significance as well as biostratigraphic utility. *Polycope* is
clearly the dominant genus group from sediment dated ~40-12 ka in 32-MC/GC
and all sites on the Lomonosov and Mendeleev Ridges (Figs. 4, 5), signifying high
productivity likely due to an intermittent, rapidly oscillating sea-ice edge at the
surface. *P. caudata* has varying percentages (3-14%) in sediment dated ~40-12
ka, depending on the site. *P. caudata* is an indicator of AI water and Cronin et al.
(2014) report that it appears to be ecologically linked to the surface conditions.
*Cytheropteron* spp. is present in moderate abundance (20-30%) in sediment dated
~35-15 ka.
Overall, the faunal characteristics from this time period imply relatively restricted
and/or poorly ventilated intermediate waters near the 32-MC/GC site. The major
exception to this corresponds with the pronounced peaks in *Krithe* and *R. mirabilis*.
This significant shift in faunal composition implies changes in ice margins, AW
inflow, deep ocean ventilation and/or enhanced deep-water transfer between the
Central Arctic Ocean and the North Atlantic.
4.4 The Last Deglacial Interval (~15 to 11 ka)
The major shift from *Polycope*-dominated to *Cytheropteron*-*Krithe*-dominated
assemblages occurs in sediment dated 12 ka in 32-MC/GC and ~15-12 ka in other
Lomonosov and Mendeleev Ridge cores. In 32-MC/GC, *Krithe* reappears in low
(10%) but significant abundance after 11 ka after being absent during MIS 2. Both
*Cytheropteron* and *Krithe* are typical faunas in NADW. Although low sedimentation
rates prevent precise dating of this shift, it almost certainly began ~14.5 ka at the
Bølling-Allerød warming transition. Because the Bering Strait had not opened yet
(Jakobsson et al., 2017), this faunal shift must have been related to one or several
of the following changes: (1) atmospheric warming; (2) strong Atlantic Water inflow
through the Barents Sea; and (3) strong Atlantic Water inflow through the eastern
Fram Strait. *A. arcticum* is absent or rare (<2% of the assemblage) in sediment
dated ~15-12 ka, suggesting minimal perennial sea ice cover and probably
summer sea-ice free conditions during late deglacial warming.
4.5 The Holocene (~11 to Present)
*Krithe* and *Cytheropteron* remain abundant in sediment dated ~10-7 ka (early
Holocene) across most of the central Arctic Basin, signifying continued influence of
water derived from the North Atlantic Ocean (Figs. 4, 5). Also during this time, *R.*
*mirabilis* reappears and spikes to an abundance of 55 percent at ~8 ka. *A.*
*arcticum* (which represents the *A. arcticum* zone) increases to >6-8% abundance
beginning in sediment dated ~7 ka, and increases to >10% abundance in sediment
dated ~3 ka. This increase in abundance is correlated with an increase in
perennial-sea ice, and is more prominent in cores from the Lomonosov Ridge than
in cores from the Mendeleev Ridge (most likely due to more persistent perennial
sea ice cover over the Lomonosov Ridge sites). The inferred middle to late
Holocene development of perennial sea ice is consistent with interpretations from
other sea-ice proxies (Xiao et al., 2015) and with the transition from an early-
middle Holocene "thermal maximum" (Kaufman et al., 2004, 2016) to cooler
conditions during the last few thousand years.
4.6 *Rabilimis mirabilis:* New faunal events signifying rapid oceanographic change
In addition to the standard ostracode zones discussed above, the cores from the
SWERUS 2014 expedition provide evidence of uncharacteristic and brief, yet
significant events of faunal dominance of a taxon.  Such events are indicative of
rapid environmental change. For example, prior studies have documented range
shifts in Arctic benthic foraminifera during the last deglacial and Holocene intervals
from the eastern Arctic Ocean (Wollenburg et al., 2001), the Laptev Sea
(Taldenkova et al., 2008, 2012), the Beaufort Sea and Amundsen Gulf (Scott et al.,
2009) and in older sediments (Polyak et al., 1986, 2004; Ishman et al., 1996;
Cronin et al., 2014). The SWERUS-32 data reveal two *Rabilimis mirabilis* "events"
-- intervals containing high proportions of this shallow water ostracode species
dated at ~45-36 ka and 9-8 ka. The modern circum-Arctic distribution of *R.*
*mirabilis* is confined to shallow (<200 m) water depths (Fig. 6a, b, and c; Hazel,
1970; Neale and Howe, 1975; Taldenkova et al., 2005; Stepanova, 2006; Gemery
et al., 2015). *R. mirabilis* can also tolerate a range of salinities, explaining its
presence in regions near river mouths with reduced salinity (Fig. 6a). *R. mirabilis*
also occurs in 2014 SWERUS-C3 multicore top samples on the Eastern Siberian
Sea slope (Supplementary Table S5; cores 23-MC4 (4%, 522 m); 18-MC4 (18%,
349 m); 16-MC4 (11%, 1023 m); 15-MC4 (41%, 501 m) and 14-MC4 (70%, 837
m). These locations correspond to the summer sea-ice edge that has receded
during recent decades over the Lomonosov Ridge.
Figures 7a and 7b show the stratigraphic distribution of *R. mirabilis* at the new
SWERUS site and other sites on the Lomonosov Ridge (96-12-1PC), the
Mendeleev Ridge (P1-94-AR-PC10) and Northwind Ridge (P1-92-AR-PC40) and
in longer cores on the Lomonosov and Northwind Ridge. These patterns suggest a
depth range extension of *R. mirabilis* into deeper water (700 to 1673 m) during
interstadial periods (MIS 5c, 5a, 3). The abundance of *R. mirabilis'* reaches 40-
50% of the total assemblage at Lomonosov Ridge site 96-12-1PC at a water depth
of 1003 m. Such anomalously high percentages of well-preserved adult and
juvenile specimens of *R. mirabilis* indicate that they were not brought to the site
through sediment transport from the shelf. Instead, the *R. mirabilis* events
represent in-situ populations. Although these *R. mirabilis* events are not
synchronous, most occur in sediment dated ~96-71 ka (late MIS 5) and at
SWERUS-C3 sites of 32-MC and 32-GC in sediment dated 45-36 ka and ~9-8 ka
(early Holocene). Thus the *R. mirabilis* events correlate with interglacial/interstadial
periods that experienced summer sea-ice free and/or sea-ice edge environments
where there may have been enhanced flux of surface-to-bottom organic matter.
However, additional study of cores from Arctic margins will be required to confirm
the paleoceanographic significance of *R. mirabilis* migration events.
5. Conclusions
Changes in ostracode assemblages in new cores from the central Arctic Ocean
signify major paleoceanographic shifts at orbital and suborbital scales during the
last 50 ka. Peaks in dominant ostracode taxa include: (1) *Krithe* zone (~45-35 ka);
(2) *Polycope* zone (~40-12 ka); (3) *Cytheropteron-Krithe* zone (~12-7 ka); and (4)
*Acetabulastoma arcticum* zone (~7 ka-present). Brief yet significant depth
migrations of *R. mirabilis* corresponding with the *Krithe* zone and *Cytheropteron-*
*Krithe* zone imply rapid paleoceanographic changes associated with influx of
Atlantic Water and/or deep ocean convection during suborbital events in MIS 3 and
the late deglacial to early Holocene. When ostracode assemblage patterns in 32-
MC/GC cores are compared to similar records from the Northwind, central
Lomonosov, Mendeleev and Gakkel Ridges (Cronin et al., 1995, 2010, Poirier et
al., 2012), these changes demonstrate pan-Arctic, nearly synchronous changes in
benthic ecosystems in association with rapid sea ice, surface productivity, and
oceanographic changes in the Atlantic Water and Arctic Intermediate Water during
MIS 3-1 (the last 50 ka). These results confirm the sensitivity of Arctic benthic
fauna to large, sometimes abrupt, climate transitions.

Acknowledgements
We are grateful to the crew of Icebreaker *Oden* and the SWERUS-C3 Scientific
Team. Thanks to A. Ruefer for assistance with sample processing. This
manuscript benefitted from reviews by X. Crosta, A. de Vernal, C. Swezey and M.
Toomey. This study was funded by the U.S. Geological Survey Climate R&D
Program. Any use of trade, firm, or product names is for descriptive purposes only
and does not imply endorsement by the U.S. Government. A. Koshurnikov
acknowledges financial support from the Russian Government (grant number 14,
Z50.31.0012/03.19.2014). Data presented in the article acquired during the 2014
SWERUS-C3 expedition are available through the Bolin Centre for Climate
Research database: http://bolin.su.se/data/.
Fig 1. a.) International Bathymetric Chart of the Arctic Ocean showing the location
of this study's primary sediment cores on the Lomonosov Ridge (red star: 32-GC2
and 32-MC4), and other core sites discussed in this paper (black circles, white
circles). (See Table 1 for supplemental core data.) White circles designate cores
that contain *Rabilimis mirabilis* events. Red arrows show generalized circulation
patterns of warm Atlantic water in the Arctic Ocean. White arrows indicate the
surface flow of the Transpolar Drift, which moves sea ice from the Siberian coast
of Russia across the Arctic basin, exiting into the North Atlantic off the east coast
of Greenland. Transect line through the map from "1" in the Chukchi Sea to "2" in
the Barents Sea shows direction of temperature profile in Fig1b.
b.) Cross section of modern Arctic Ocean temperature profile from showing major
water masses. PSW: polar surface water, AL: Atlantic layer, AIW: Arctic
intermediate Water, AODW: Arctic Ocean Deep water. Ocean Data View Source:
Schlitzer, 2012. Ocean Data View: http//odv.awi.de
Fig. 2 Chronology and stratigraphy of SWERUS-32-GC and 32-MC. Bulk density
and magnetic susceptibility profiles for 32GC were previously correlated to the
well-dated 96-12-1PC core by Jakobsson et al. (2016). Bulk density primarily
reflects changes in grain size, with coarser material having a higher density than
finer grained material. The overall position of MIS 5 is supported by the occurrence
of *E. huxleyi*. The chronology for the upper 30-35 cm is based on radiocarbon
dating in both 32-MC and 32-GC. Beyond the range of radiocarbon dating, an
extrapolation to the inferred position of MIS 3/4 boundary (57 ka at 105 cm) is
applied.
Fig 3. Relative frequencies (percent abundance) of dominant taxa in SWERUS-C3
32-MC and 32-GC. The y-axis shows the modeled, mean age during a 2-sigma
range of uncertainty.
Fig 4. Relative frequencies (percent abundance) of dominant taxa in SWERUS
32-MC (dotted line) compared to other Lomonosov Ridge cores 2185, 2179 and
AOS94 28 (Poirier et al., 2012). The chronology for core PS 2185-4 MC (1051 m)
is described in Jakobsson et al., 2000, Nørgaard-Pederson et al., 2003,
Spielhagen et al., 2004; core PS 2179-3 MC (1228 m) in Nørgaard-Pederson et
al., 2003 and Poirier et al., 2012; and core AOS94 28 (PI-94-AR-BC28, 1990 m) in
Darby et al., 1997.

Fig 5.  Relative frequencies (percent abundance) of dominant taxa in SWERUS 32-MC (dotted line) compared to other Mendeleev Ridge cores AOS94 8 (Poirier et al., 2012), AOS94 12, and HLY6. The chronology for core HLY6 (HLY0503-06JPC, 800 m) is described in Cronin et al., 2013; core AOS94 8 (PI-94-AR-BC8, 1031 m) in Cronin et al., 2010 and Poirier et al., 2012; and core AOS94 12A (PI-94-AR-BC12A, 1683 m) in Cronin et al., 2010.

Fig 6.  a.) Occurrence map of *Rabilimis mirabilis* in the Arctic Ocean and surrounding seas based on 1340 modern surface samples in the Arctic Ostracode Database (AOD; Gemery et al., 2015).

b.) Modern depth and c.) latitudinal distribution of *R. mirabilis* based on 1340-modern surface samples in the AOD (Gemery et al., 2015).

Fig 7. a.) Relative frequency (percent abundance) of *R. mirabilis* in SWERUS-32 cores and in central Arctic Ocean cores, 160 ka to present.  b.) *R. mirabilis* in core LOMROG07-04 from 260 ka to present and in core P1-92-AR-PC30 from 340 ka to present.

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

Table 1. Expedition and core site data for cores presented in this study.

| Year | Expedition | Core name | Latitude | Longitude | Water depth (m) | Location |
|------|-----------|-----------|----------|-----------|-----------------|----------|
| 2014 | SWERUS-L2 | SWERUS-L2-32-MC4 | 85.14 | 151.59 | 837 | Lomonosov Ridge |
| 2014 | SWERUS-L2 | SWERUS-L2-32-GC2 | 85.15 | 151.66 | 828 | Lomonosov Ridge |
| 2014 | SWERUS-L2 | SWERUS-L2-24-MC4 | 78.80 | 165.38 | 982 | E. Siberian Sea Slope |
| 2014 | SWERUS-L2 | SWERUS-L2-28-MC1 | 79.92 | 154.35 | 1145 | E. Siberian Sea Slope |
| 2014 | SWERUS-L2 | SWERUS-L2-33-TWC1 | 84.28 | 148.65 | 888 | Lomonosov Ridge |
| 2014 | SWERUS-L2 | SWERUS-L2-34-MC4 | 84.28 | 148.71 | 886 | Lomonosov Ridge |
| 1994 | AOS SR96-1994 | PI-94-AR-BC28 | 88.87 | 140.18 | 1990 | Lomonosov Ridge |
| 1991 | Arctic 91 | PS 2179-3 MC | 87.75 | 138.16 | 1228 | Lomonosov Ridge |
| 1991 | Arctic 91 | PS 2185-4 MC | 87.53 | 144.48 | 1051 | Lomonosov Ridge |
| 1994 | AOS SR96-1994 | PI-94-AR-BC8 | 78.13 | 176.75 | 1031 | Mendeleev Ridge |
| 1994 | AOS SR96-1994 | PI-94-AR-BC12A | 79.99 | 174.29 | 1683 | Mendeleev Ridge |
| 2005 | HOTRAX | HLY0503-6 | 78.29 | -176.99 | 800 | Mendeleev Ridge |
| 1994 | AOS SR96-1994 | P1-94-AR-PC10 | 78.15 | -174.63 | 1673 | Mendeleev Ridge |
| 1992 | USGS-Polar Star | P1-92-AR P40 | 76.26 | -157.55 | 700 | Northwind Ridge |
| 1992 | USGS-Polar Star | P1-92-AR-P30 | 75.31 | -158.05 | 765 | Northwind Ridge |
| 2007 | LOMROG 07 | LOMROG07-PC-04 | 86.70 | -53.77 | 811 | Lomonosov Ridge |
| 1996 | Oden 96 | 96-12-1PC | 87.10 | 144.77 | 1003 | Lomonosov Ridge |

Table 2. Radiocarbon dates for SWERUS 32 cores, uncalibrated $^{14}$C age and calibrated $^{14}$C chronology.

| 32-MC/GC chronology | | Unmodelled 2 sigma (2 std dev) | | | | Modelled 2 sigma (2 std dev) | | | |
|---------------------|-------|------|------|------|-------|------|------|--------|-------|
| Lab number ($^{14}$C date age, error) | Depth (cm) | from | to | mean | errror | from | to | mean** | error |
| OS-124799 (3410, 25) | 2.5 | 3168 | 2698 | 2912 | 124 | 3045 | 2605 | 2802 | 105 |
| OS-124798 (6110, 20) | 4.5 | 6435 | 5974 | 6213 | 116 | 6317 | 5902 | 6140 | 113 |
| OS-124599 (7920, 35) | 5.5 | 8313 | 7874 | 8085 | 110 | 8176 | 7766 | 7972 | 101 |
| OS-124598 (8290, 30) | 8.5 | 8715 | 8207 | 8465 | 119 | 8576 | 8187 | 8385 | 99 |
| OS-124597 (11000, 35) | 11.5 | 12525 | 11661 | 12094 | 222 | 12191 | 11353 | 11831 | 230 |
| OS-124754 (11200, 40) | 14.5 | 12635 | 12040 | 12365 | 164 | 12625 | 12008 | 12381 | 165 |

| | | | | | | | | |
|---|---|---|---|---|---|---|---|---|
| OS-125185 (18650, 80) | 19.5 | 22116 | 21357 | 21729 | 183 | 21973 | 21252 | 21637 | 179 |
| OS-125190 (29400, 280) | 24.5 | 33567 | 31805 | 32733 | 462 | 33298 | 31585 | 32423 | 455 |
| OS-125192 (35400, 560) | 31.5 | 40705 | 38099 | 39301 | 638 | 40858 | 38451 | 39608 | 608 |
| OS-127484 (40000, 1700) | 33* | 47589 | 40881 | 43837 | 1646 | 44472 | 40403 | 42428 | 1002 |

All ages as calibrated years BP

$\Delta R$ = 300 ±100 years (Reimer and Reimer, 2001)
Marine13 calibration curve (Reimer et al., 2013)
*Sample collected from 32-GC, original depth was 36 cm but corrected by 3 cm based on ostracode correlation with 32-MC
**We used the modeled, mean, 2-sigma age to plot species' relative frequencies.

Table 3. List of species included in genus-level groups.

| Group name | Species included in Group: |
|---|---|
| *Cytheropteron* spp. | *Cytheropteron higashikawai* Ishizaki 1981, *Cytheropteron scoresbyi* Whatley and Eynon 1996, *Cytheropteron sedovi* Schneider 1969 and *Cytheropteron parahamatum* Yasuhara, Stepanova, Okahashi, Cronin and Brouwers 2014. |
| *Polycope* spp. | *P. inornata* Joy & Clark, 1977 and *P. bireticulata* Joy & Clark, 1977. (For scanning electron microscope images, see Joy & Clark 1977: Plate 3, Fig. 1; Yasuhara et al., 2014: Plate 3, Figs 2-5 and Plate 2, Figs 1-2.) May include: *P. arcys* (Joy & Clark, 1977), *P. punctata* (Sars 1869), *P. bispinosa* Joy & Clark 1977, *P. horrida* Joy & Clark 1977, *P. moenia* Joy & Clark 1977, *P. semipunctata* Joy & Clark 1977, *P. obicularis* Sars 1866. *P. pseudoinornata* Chavtur, 1983 and *P. reticulata* Muller 1894 |

Table 4. Summary of indicator species, pertinent aspects of their modern ecology and paleoenvironmental significance.

| Species | Modern ecology / paleoenvironmental significance |
|---|---|
| *Acetabulastoma arcticum* (Schornikov, 1970) | The stratigraphic distribution of *A. arcticum* is used as an indicator of periods when the Arctic Ocean experienced thicker sea-ice conditions but not fully glacial conditions when productivity would have halted. This pelagic ostracode is a parasite on *Gammarus* amphipods that live under sea ice in modern, perennially sea-ice-covered regions in the Arctic (Schornikov, 1970). Cronin et al. (2010) used *A. arcticum's* presence in 49 late Quaternary Arctic sediment cores as a proxy to reconstruct the Arctic Ocean's sea-ice history during the last ~45 ka. |
| *Krithe* spp. | Species of the genus *Krithe* typically occur in low-nutrient habitats spanning across a range of cold, interstadial temperatures but are especially characteristic of AODW (Cronin et al., 1994; 1995; 2014). In SWERUS-32 cores, *K. hunti* was far more prevalent than *K. minima*. From a modern depth-distribution analysis using AOD, *K. hunti* appears in greatest abundance (50-80% of the assemblage) at depths between 2000-4400 mwd, however, this taxon is also found in significant numbers (20-50%) at depths between 400-2000 m. With a preference for deeper, cold, well-ventilated depths, *Krithe spp.* events are useful in identifying late Quaternary shifts in Arctic Ocean water masses and making biostratigraphic correlations (Cronin et al., 2014). |
| *Polycope* spp. | Today, this Atlantic-derived, myodocopid genus is in highest abundance (40-60% of assemblage) in cold intermediate-depth waters between 800-2300 mwd. It characterizes fine-grained, organic rich sediment in well-oxygenated water. In fossil assemblages, *Polycope* is indicative of areas with high productivity that are seasonally ice-free or have variable or thin sea-ice cover (Cronin et al., 1995; Poirier et al., 2012). |
| *Cytheropteron* spp. | The two dominant *Cytheropteron* species in 32-MC and 32-GC are *C. sedovi* and *C. scoresbyi,* along with lower but significant numbers of *C. parahamatum* (reaches 24% of assemblage at 10 ka) and *C. higashikawai* (fluctuates in very low numbers between 0-3% at any given time in downcore samples). These particular *Cytheropteron* species are broadly diagnostic of deeper, well-ventilated water masses (AIW and AODW). |
| *Pseudocythere caudata* Sars 1866 | This species of N. Atlantic origin rarely exceeds >15% in modern Arctic Ocean assemblages. It characterizes lower AW and AI water at depths of 1000-2500 mwd. It usually co-occurs with *Polycope* spp. in fossil assemblages and may be associated with surface conditions (Cronin et al., 1994, 1995, 2014), but more work needs to be done on its ecological significance. |

Table 5. Although *R. mirabilis* (Brady, 1868) is known and named from Pleistocene sediments in England and Scotland (Brady et al., 1874), this list cites various workers since that have documented this species in Arctic deposits dating back to the late Pliocene, when summer bottom temperatures were inferred to be up to 4°C warmer than today.

| Citation | Location / Formation (Age) |
|---|---|
| Siddiqui (1988) | Eastern Beaufort Sea's Iperk sequence (Plio-Pleistocene) |
| Repenning et al. (1987) | Alaska's North Slope Gubik Formation (Pliocene) |
| Penney (1990) | Central North Sea deposits (early Pleistocene age,1.0-0.73 Ma) |
| Feyling-Hassen (1990) | East Greenland's Kap København Formation (late Pliocene) |
| Penney (1993) | East Greenland's Lodin Elv Formation (late Pliocene) |

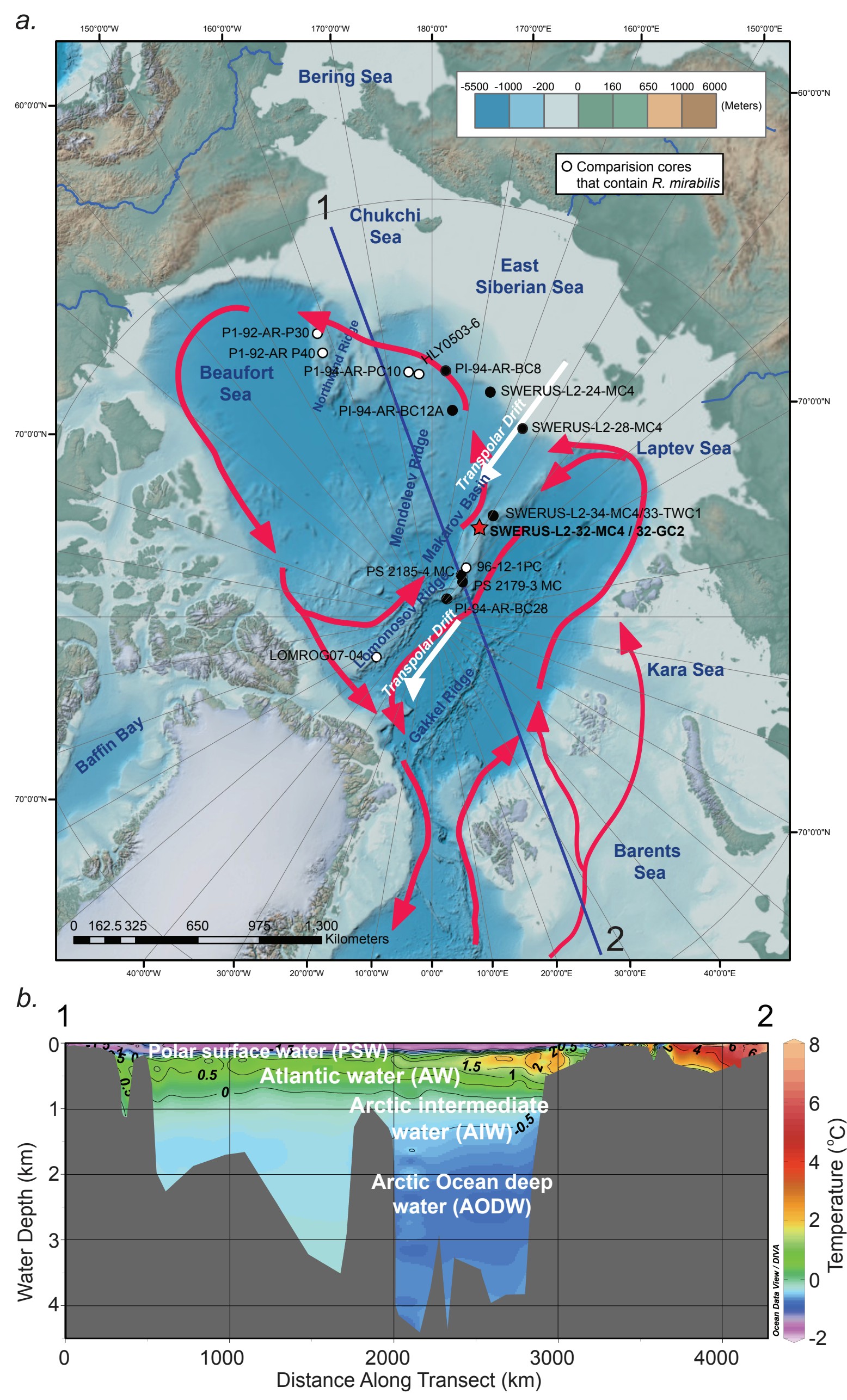

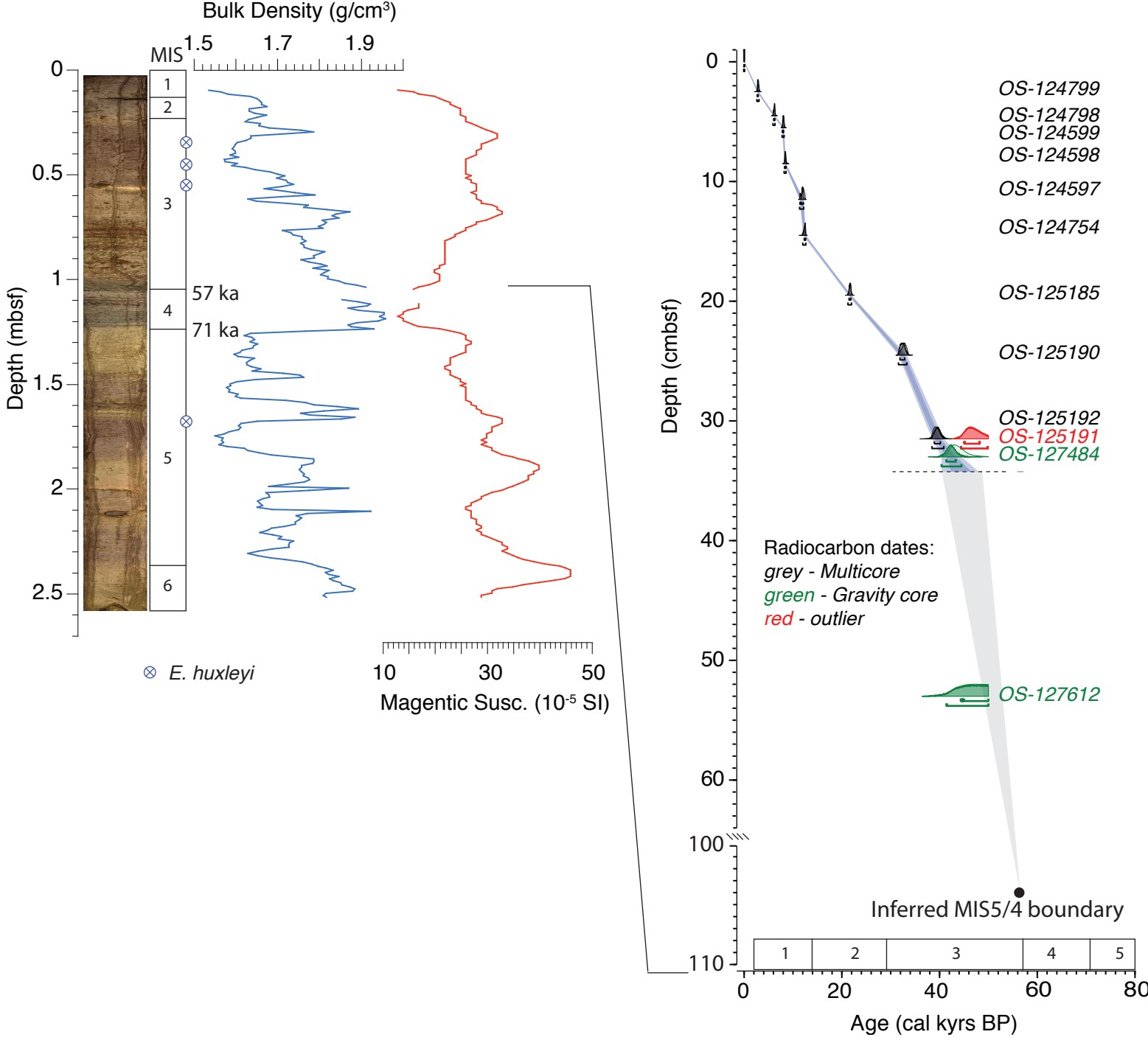

Bulk Density (g/cm³)

MIS

*E. huxleyi*

Magentic Susc. (10⁻⁵ SI)

57 ka

71 ka

Radiocarbon dates:
*grey - Multicore*
*green - Gravity core*
*red - outlier*

OS-124799
OS-124798
OS-124599
OS-124598
OS-124597
OS-124754
OS-125185
OS-125190
OS-125192
*OS-125191*
*OS-127484*
*OS-127612*

Inferred MIS5/4 boundary

Age (cal kyrs BP)

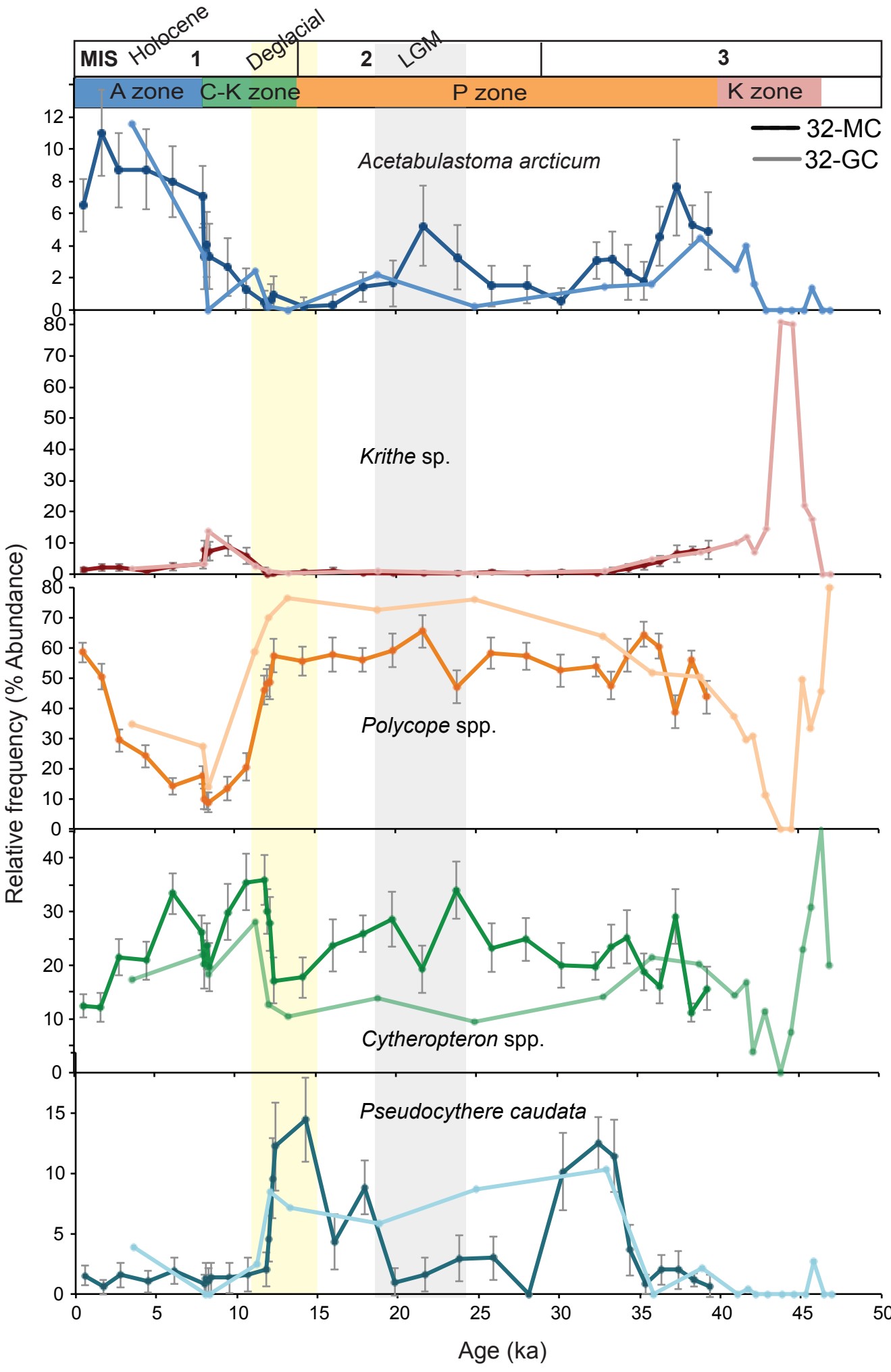

# Lomonosov Ridge

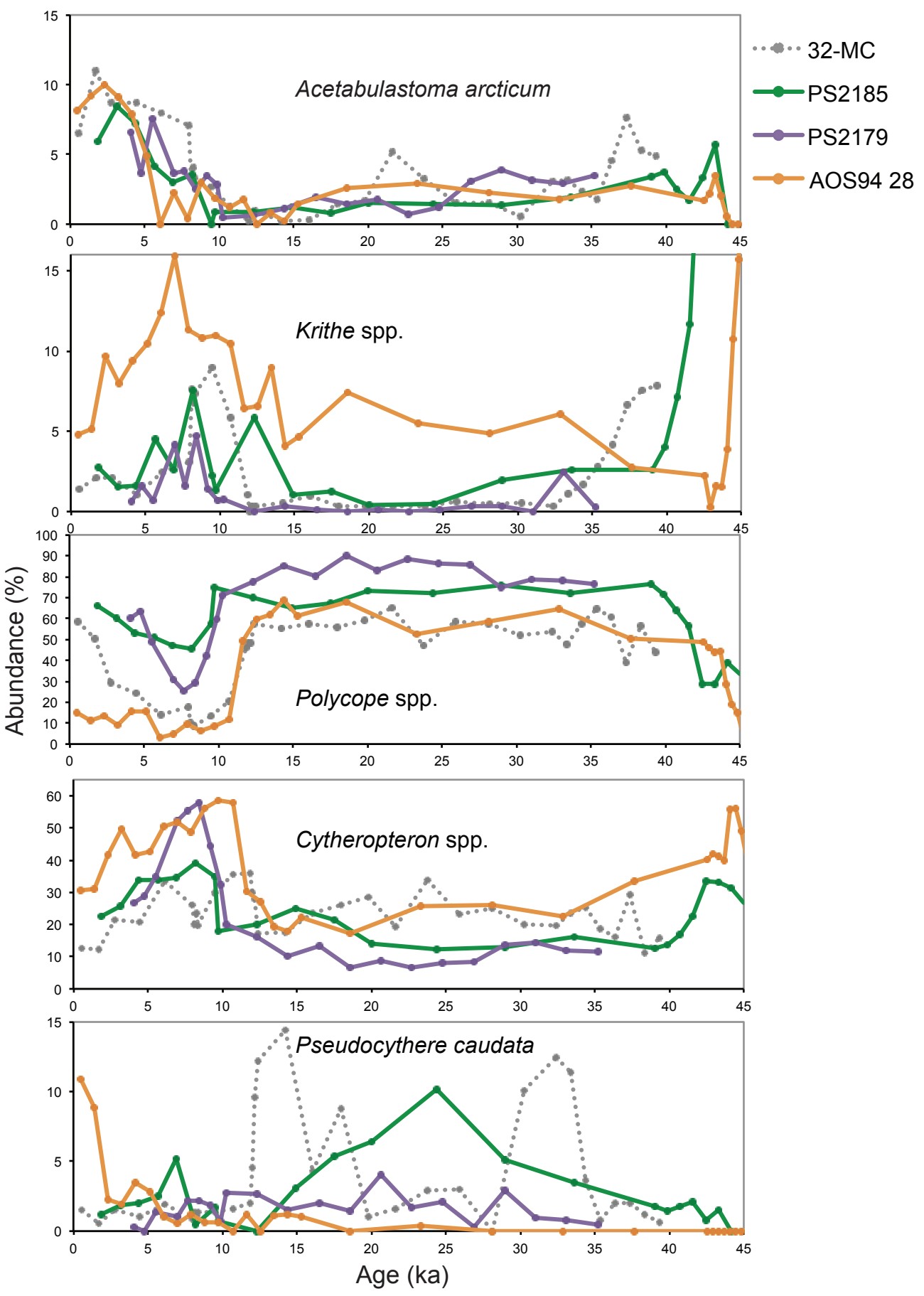

# Mendeleev Ridge

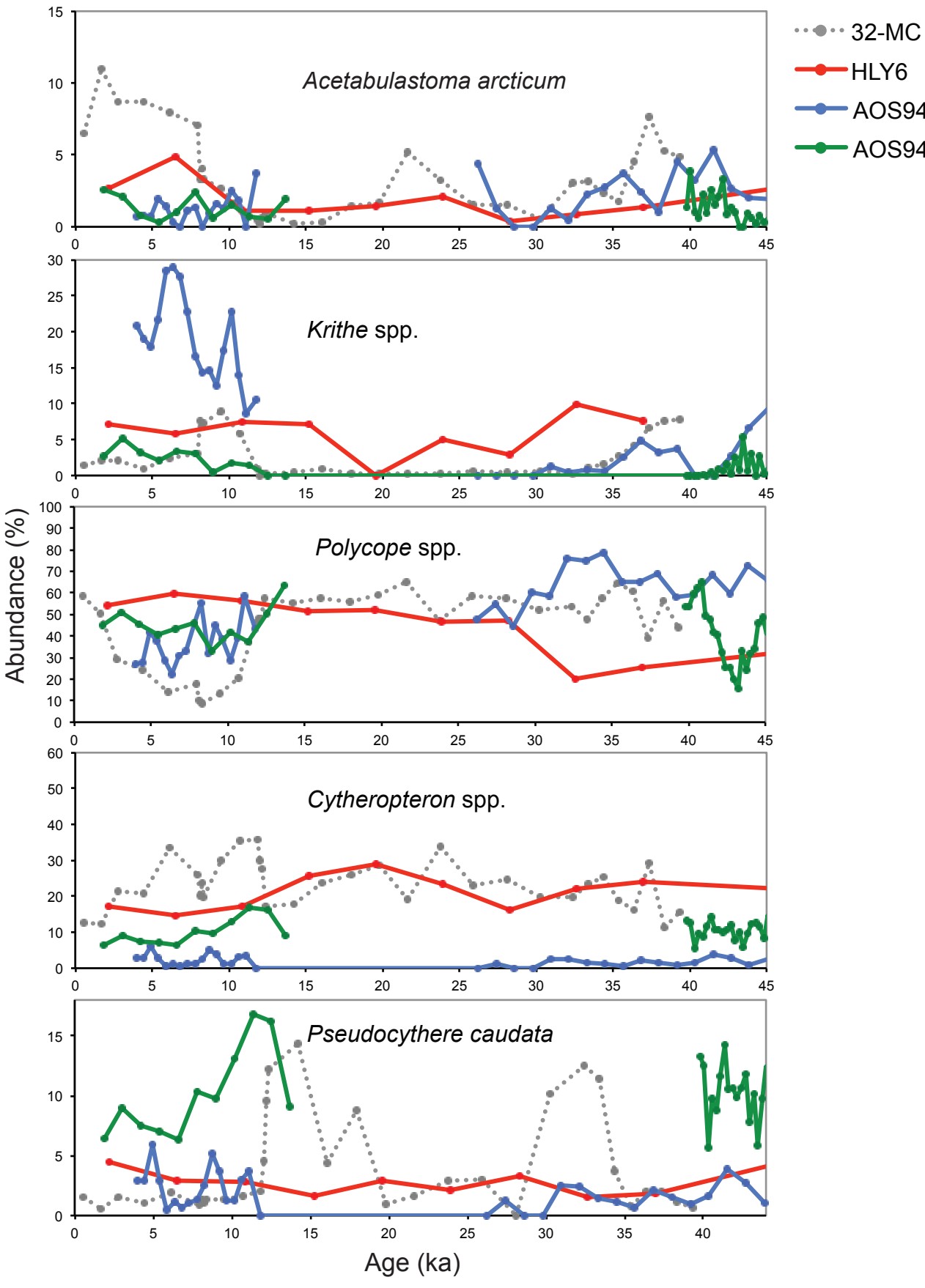

*a.*

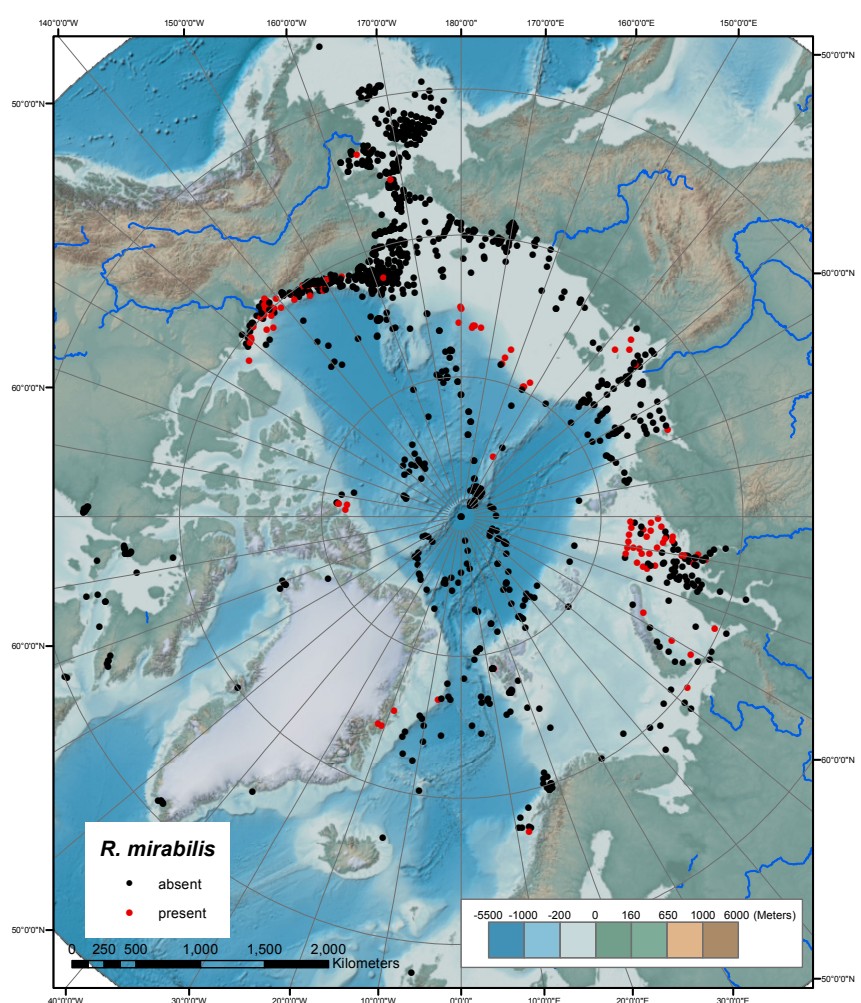

*R. mirabilis* modern depth and latitude distribution

*b.*

Number of specimens in AOD sample (n=115)

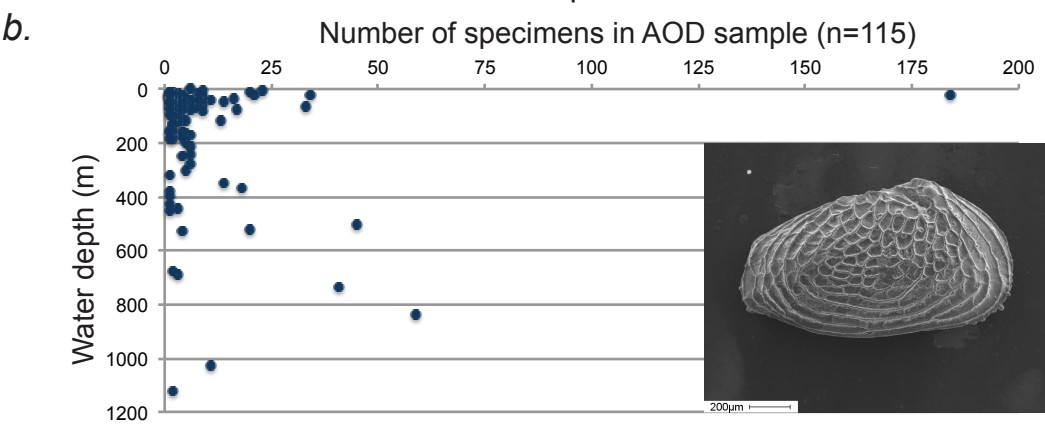

*c.*

Percent abundance in AOD sample (n=115)

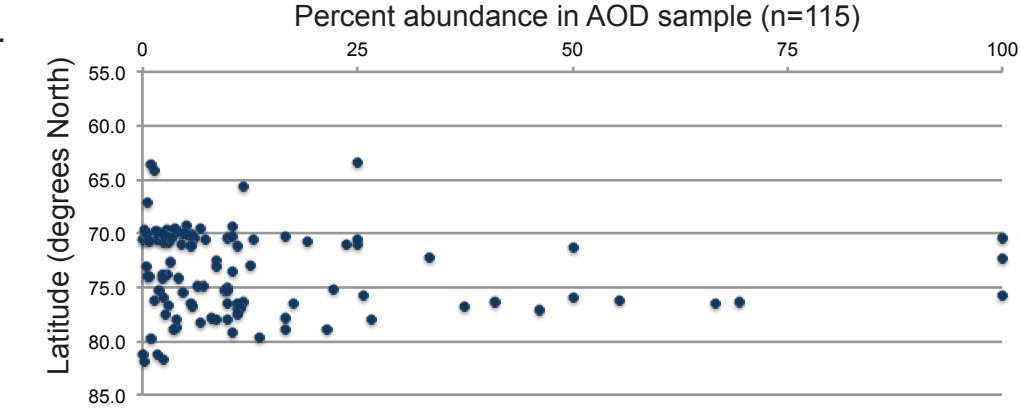

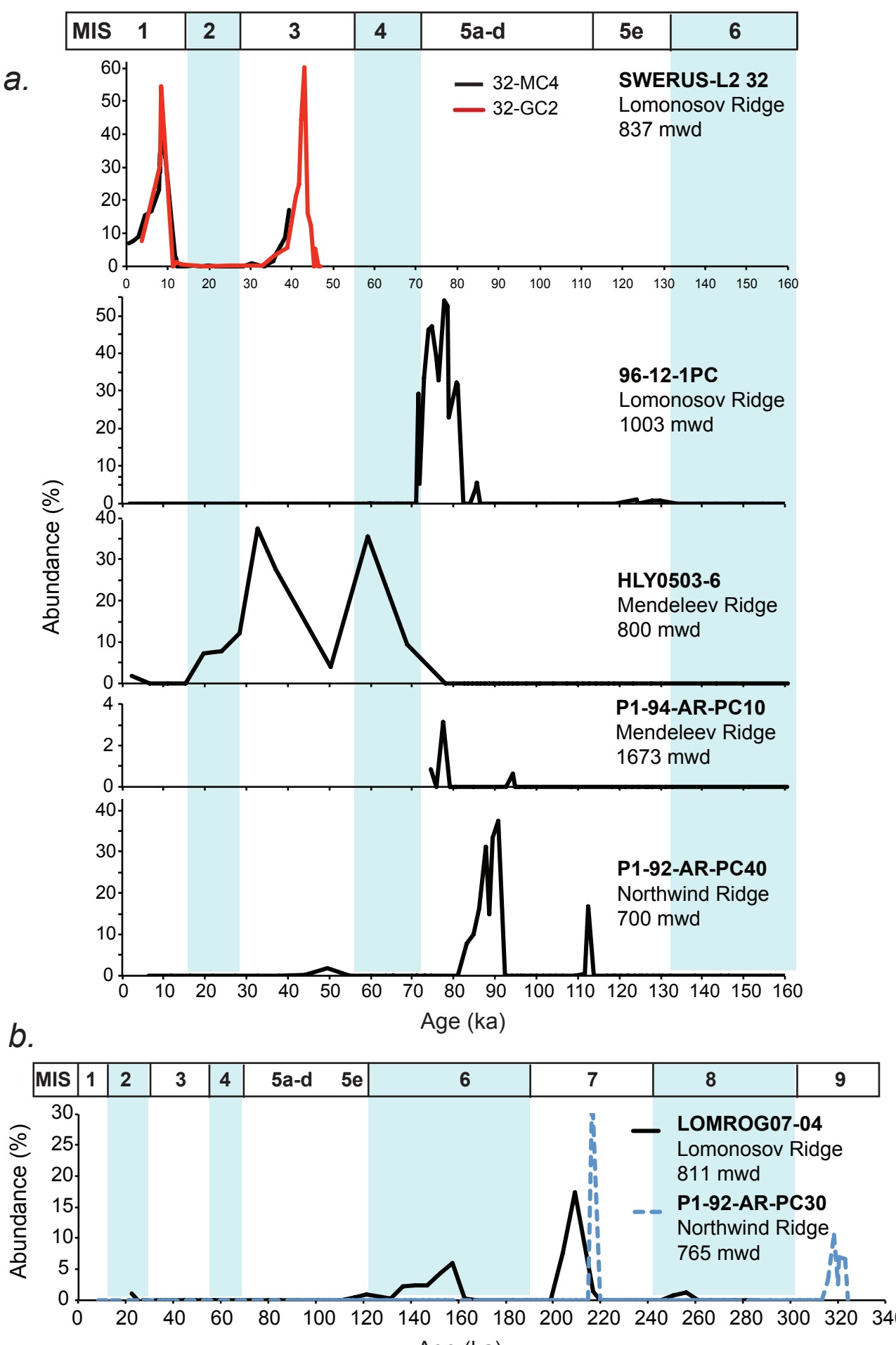

*R. mirabilis* events