# Peer review of "Central Arctic Ocean paleoceanography from ~50 ka to present, on the basis of ostracode faunal assemblages from SWERUS 2014 expedition"

_Climate of the Past, 2017_

## Referee Comment (RC1) · X. Crosta (Referee) · 7 Apr 2017

In the context of global warming and recent Arctic sea ice waning, it is important to understand the natural forcing of past sea ice changes. Here, Gemery and co-authors present a low resolution reconstruction of Central Arctic sea ice changes over the past 50,000 years using ostracode faunal assemblages in two twin cores retrieved in 2014. Although such records are highly necessary, the manuscript suffers from several limitations and flaws that prevent acceptation in its present form. First, the manuscript does not go further than the previous study published by the same group (Cronin et al., 2010) in which conclusions were exactly the same. Central Arctic sea ice was reconstructed in several cores from the Lomonosov Ridge, over the same time period. It

was evidenced that "Results suggest intermittently high levels of perennial sea ice in the central Arctic Ocean during Marine Isotope Stage (MIS) 3 (25-45 ka), minimal sea ice during the last deglacial (16-11 ka) and early Holocene thermal maximum (11-5 ka) and increasing sea ice during the mid-to-late Holocene (5-0 ka)". Similar interpretations are here presented by Gemery and co-atuhors. The only addition to Cronin et al. (2010) is that "sea-ice cover during the last glacial maximum may have been less extensive at the southern Lomonosov Ridge at our core site ($\sim$85.15°N, 152°E) than farther north and towards Greenland", which is pretty weak.

Second, the manuscript is only descriptive and does not present any forcing mechanisms to explain the observed changes in sea ice cover over the past 50,000 years. Why the MIS 3 did not experience perennial sea ice cover when temperatures where globally lower than during the Late Holocene? What is the link between intermittent perennial and seasonally ice-free conditions during MIS3 and HE/DO? What is the impact of lower sea-level during MIS3 on ocean circulation (less to no North Pacific waters), on sea ice formation (mainly on marginal seas if I am right) and sea ice transport off the Arctic Ocean? The new data should be presented and explained in the context of large scale ocean and atmosphere changes over the past 50,000 years. There are plenty of publications from the GIN Seas and Fram Strait to document NADW inflow (marked here by Krithe spp. and Cytheropteron spp.) and AW outflow (marked here by Polycope spp. and P. caudata). There is also a wealth of publications from continental peri-Arctic to document atmospheric patterns and their impact on central Arctic sea ice. As such, the very attractive title is misleading.

Third, results are discussed in "climatic phases" that are not congruent with the ostracode faunal changes. It is more sensible to discuss changes in the four "ostracode zones". I however do not fully agree on the four zones. Based on faunal changes more periods can be discussed: The K zone, a first increase in A. arcticum between 42-35 kyrs BPP, a P. caudata peak between 35-27 kyrs BP, a second increase of A. arcticum between 25-20 kyrs BP, a second P. caudata peak between 20-12 kyrs BP,

the C zone and the A zone. There is no information on why there are so much difference in ostracode abundances and species numbers between the twin cores. Line 266-271: The shift between Polycope spp. and the Krithe-Cytheropteron group is at 12 kyrs BP not 14.5 kyrs BP. And the Krithe gp is less than 10%. Is this small increase significant? Over the deglaciation I see the following sequence: P. caudata (20-12 kyrs BP); Cytheropteron (12-9 kyrs BP); Krithe (10-7 kyrs BP). This is not really discussed. Line 280: Krithe spp. are less than 10%. This is not what I call abundant.

Fourth, the "Results" part present description of results mingled with some environmental interpretations. And the "Discussion" part does not present any environmental interpretations nor forcing mechanisms. The structure should be modified accordingly. Lines 307-325: Useless in the paper. Authors should stick to paleoceanographic reconstructions and interpretations.

Fifth, the paper oscillates between presenting new sea ice reconstructions (but no explanation of such changes) and validation of R. mirabilis to infer past sea ice changes. I would say that these are two different topics mand should be presented in two different papers. Additionally, records of R. mirabilis should be described in the "Results" part. They here appear out of the blue at the very end of paper. Lines 328-330: Ostracode species mentioned here are not presented in the results. There is no way to compare and assess what is written. Although it is difficult to assess here because the records are presented in different plots, it seems to me that R. mirabilis record in the twin cores are similar to the Krithe spp. record with peaks centered at 42-44 kyrs BP and 10-5 ka BP. This contradicts lines 328-333 where authors state that R. mirabilis modern distribution mimics B aculeata's one. This should be expanded. Why these two species share a similar modern distribution (linked to perennial sea ice) while presenting different down-core records whereby B. aculeata is still linked to perennial sea ice while R. mirabilis goes together with species tracking less sea ice and NADW influx into central Arctic?

Sixth, the "Chronology" part is not totally clear to me. Data used to estimate the mentioned 3cm offset between the MC and GC cores are not presented. The tuning below 31.5 cm is not presented. It seems that there is only one point with E. huxleyi to infer the MIS5. I strongly doubt that the mean reservoir age was constant through time. It should be acknowledges even though this may not have a big impact on the results/interpretations here due to low temporal resolution.

Seventh, the "Introduction" is very weak. The scientific issue is not very well presented (only in first and last paragraph). There is not state-of-the art. I suggest to much better highlight the difference to Cronin et al. (2010).

————————————————————

---

## Referee Comment (RC2) · A. de Vernal (Referee) · 4 Jun 2017

A. de Vernal (Referee)

devernal.anne@uqam.ca

The manuscript by Gemery et al. addresses an important topic, that of the ocean and climate change in the Arctic during the Quaternary. The new data from the SWERUS core 32 add useful information on the stratigraphy of ostracods over the last 40,000 years in the Arctic Ocean. The study core is one of the rare relatively well-dated sequence from the central Arctic Ocean, at least for the last 35 kyr and relatively high sedimentation rates (∼ 1 cm/kyr on average) permit to report the stratigraphical distribution of microfossils with millennial time resolution.

The new results from core 32 are very interesting. They are used together with the

data from many other cores (most being already published) to present an Arctic Ocean wide synthesis for the last ∼ 40 kyr. This offers a very valuable contribution as announced in the title and summarized in the abstract. In the manuscript, however, other data encompassing longer time scales, ranging up to the 160 kyr or even 340 kyrs, are discussed with reference to occurrence peaks of Rabimilis mirabilis in the ostracode assemblages. Hence, the scope of the paper is not clear. There is a hiatus between the abstract summarizing the new data from the SWERUS core 32 data and the discussion dealing with the longer time scales. In my opinion, the new data unquestionably deserve publication after a few points is clarified. The comparison with other records encompassing the last 40 kyr is very interesting and could be much useful especially if the basin-scale results are discussed in a more comprehensive manner. The synthesis part on the longer time scales, however, seems to be another story, which would require a better presentation/demonstration of the chronostratigraphy (including uncertainties) before to offer a robust scientific contribution.

My recommendation is therefore to revise the manuscript by focusing on the new data and their implication in term of large-scale paleoceanography at the scale of the last 40 kyrs. The manuscript will then offer an original, robust and useful contribution providing that some clarification/modification are made with regard to (a) the chronology and (b) the absolute abundance of ostracodes. (a) The age-depth relationship in cores 32MC and 32G was derived from linear interpolation between 14C dates as shown in figure 2. However, other solutions with highly variable sedimentation rates are very likely in the Arctic Ocean context. In particular, no accumulation or extremely low sedimentation rates during the last glacial maximum are recorded at many sites of the central Arctic Ocean (e.g., Norgaard-Pedersen et al. 2003; Polyak 2004; Not & Hillaire-Marcel 2010; hanslik et al. 2010). Hence, the age of ca. 20 ka in core 32MC can simply result from mixing. The use of a Bayesian approach (e.g., with the Bacon software for depth/age modelling; Blaauw & Christen, 2011) would be appropriate and could help constraining the uncertainties. Another concern comes for the old 14C ages (> 40 ka) that must be considered with caution because of potential biases due to even extremely small contamination (e.g., Hughen 2007), notably through diagenetic processes and carbonate recrystallisation (Sivan et al., 2002; Douka et al., 2010). Thus, the chronology of the lower part of the sequence, older than about 35 kyr, is equivocal because the absolute age as well as the linear interpolation can be questioned. A critical presentation of the age-depth relationships in the other cores from the Lomonosov and Mendeleev ridges (Figures 4 and 5) would be useful to give an information on the time window represented by the samples analyses, to strengthen the regional zonation proposed and to clearly demonstrate the synchroneity or time lags in the records.

(b) The results are presented in term of number of ostracod counted and percentages of main taxa. The concentration or density of ostracode valves per unit of weight (g) or unit of volume (cc) would be very useful to describe the real abundance of ostracod in sediment and to get a picture of the actual fluxes of the key species. Moreover, Rabimilis mirabilis is discussed as an important species, but its downcore distribution is not shown in figures 3-5. It should be added (% and concentration) in the diagrams of these figures.

Beyond clarification in the presentation of results, some discussion about the actual significance of the ostracodes in the sediment would be helpful, as briefly suggested below.

1. In the interpretative schemes of the result section, the ostracode assemblages are associated with water masses, some of Atlantic origin. Are the ostracodes indicative of actual conditions in bottom waters or to transport with water masses ?

2. Acetabulostoma arcticum is associated with multi-year sea-ice cover, which makes it a very important bio-indicator, actually the only one that can be used to assess "positively" on the occurrence of perennial sea ice as far as I know. The fact that it characterizes the postglacial on the Lomonosov Ridge is important, but its low occurrence during the glacial interval is equivocal. Can it relate to low general productivity due to too thick perennial ice ? Its low occurrence on the Mendeleev Ridge for most the study

interval is also intriguing.

3. Rabimilis mirabilis is mentioned as a shallow water taxon. Could it be transported from the shelf (with sea ice for ex.) ? The fact that both adult and juvenal specimens are recovered (lines 361-364) is not a very convincing argument.

4. The zonation from the Lomonosov Ridge seems relatively robust, but Krithe spp. and Pseudocythere caudata show somewhat different records in the study cores. How can the difference be interpreted ? Does the deeper location of core AOS94-28 matter ? Similar, the assemblages from the Mendeleev Ridge show differences notably with regard to Krithe spp. Pseudocythere caudata. Are the differences indicative of a regionalism ?

5. High abundance/dominance of Polycope spp. characterizes the pre-Holocene sediment of almost all cores (Figures 3-5). This is interesting as it might indicate uniform water masses from Atlantic origin in intermediate layers of the Arctic Ocean during glacial time.

Other minor comments :

- The supplementary tables are not easy to read and there are parts missing. Probably there was a problem when saving them as pdf.

- The nomenclature of cores in figures 4 and 5 is not exactly the same than in the map of figure 1, which is a little confusing.

- In figure 5, the spacing of data points from core HLY6 is so large that comparison with other cores is not very useful ; Linking the data points between ~12 ka and ~27 ka for core AOS94 8, and between ~ 13 ka and 40 ka for core AOS94 12 is inappropriate.

---

## Referee Comment (RC3) · Anonymous Referee #3 · 22 Jun 2017

The paper by Gemery and colleagues represents an interesting study that illustrates how the analysis of ostracod fauna can shed new light on the paleoceanographic changes occurred in the central Arctic Ocean during the Late Quaternary (ca. the last 50 ka). This study can be particularly welcomed by teams involved in the reconstruction of recent past sea-ice conditions and relative strength of Atlantic Water influx to the Arctic Ocean during periods of climate variations. The Authors also put into question the possibility to use peaks in ostracod species (i.e. Rabilimis mirabilis) commonly recorded at shallow water-depths (<200 m) as proxy for abrupt changes in paleoceanographic conditions. In my opinion, the most interesting elements of the paper are (i) the potential paleoceanographic significance of R. mirabilis migration events, (ii) the

comparison among relatively high-resolution ostracod data from several cores and (iii) the effort to reconstruct a robust chronological framework for the 2 studied cores (32-GC and 32-MC). However, there are some aspects, concerning the interpretation of ostracod data and text organization, that could be improved:

1. I have the impression that the paper, in its present state, doesn't fully emphasize all the new results derived by the analysis of ostracodfauna (cores 32-GC and 32-MC). In particular, the discussion section (section 5.) only focuses on the distribution of R. mirabilis, while it should also include the reconstruction of paleoceanographic conditions from ca. 50 ka to present (according to the scope and title of the paper), emphasizing the novelty in respect to previous studies undertaken in the same area (e.g.,higher sampling resolution). On the other hand, results concerning R. mirabilis distribution patterns are poorly described in section 4. Moreover, I suggest to describe in more detail the stratigraphic/temporal patterns of ostracod indicator species from the 2 new cores (section 4.2) and more clearly distinguish data interpretation from the discussion and conclusions, based on the comparision among several cores.

2. The ostracod zones could be a little bit refined, highlighting the occurrence of a "transitional" ostracod fauna zone, between ca. 42-35 ka, dominated by Polycope spp., but also characterized by remarkable percentages of A. arcticum and Krithe spp.. I think that the ostracod data (Fig. 3) show interesting faunal turnover that could be investigated in depth using a statistical approach. Did the Authors perform multivariate analysis (e.g., DCA) to improve the identification of the main faunal turnover through the core succession/time and the comprehension of the main controlling parameters? Moreover, it could be useful a more detailed explaination of the main turnover in terms of paleoenvironmental conditions: what do the two peaks in P. caudata (between ca. 35-30 ka and 20-12 ka) mean? I also wonder why the percentages of A. arcticum are higher during the mid-late Holocene in respect to the LGM.

3. The Authors state that the R. mirabilis peaks are composed by in-situ populations because of the presence of well-preserved adult and juvanile valves. I agree with the

Authors that this is a good autochthonity indicator, however I wonder if there are other data that can support this interpretation and/or other analyses can be performed to exclude the possibility of resedimentation events.

4. In Mendeleev Ridge area, the visual inspection of ostracod data seems to show a low degree of correlation among cores. Maybe, it could be useful to compare cores ostracod data (Figs. 4, 5) using statistical methods. How much the ostracod patterns are really similar as stated by the Authors (e.g., in the abstract "Comparisons with faunal records from other cores from the Mendeleev and Lomonosov Ridges suggest generally similar patterns, . . .")?

5. The construction of the age-depth model for the 2 new cores deserves a more detailed explaination and discussion. In particular, I'd like to see how ostacod data help to depth align the 2 cores.

Minor comments: a) Cytheropteron spp. should be added in the abstract along with the other ostracod indicator species. b) In the introduction, I suggest to more clearly state the aims of the paper and highlight the novelty of this study in respect to previous works dealing with ostracod fauna from nearby cores. c) An entire sub-section (5.1.) focused on foraminiferal fauna events is a little bit too much for a paper dealing with ostracodfauna. d) Paleoenvironmental changes documented by ostacodfauna should be reported in conclusions. e) Figure 3: please replace Krithe sp. with Krithe spp.

---

## Author Comment (AC1) · 27 Jun 2017

X. Crosta (Referee) x.crosta@epoc.u-bordeaux1.fr Interactive comment on Clim. Past Discuss., doi:10.5194/cp-2017-22, 2017.

 In the context of global warming and recent Arctic sea ice waning, it is important to understand the natural forcing of past sea ice changes. Here, Gemery and co-authors present a low resolution reconstruction of Central Arctic sea ice changes over the past 50,000 years using ostracode faunal

assemblages in two twin cores retrieved in 2014. Although such records are highly necessary, the manuscript suffers from several limitations and flaws that prevent acceptation in its present form. First, the manuscript does not go further than the previous study published by the same group (Cronin et al., 2010) in which conclusions were exactly the same. Central Arctic sea ice was re-constructed in several cores from the Lomonosov Ridge, over the same time period. It was evidenced that "Results suggest intermittently high levels of perennial sea ice in the central Arctic Ocean during Marine Isotope Stage (MIS) 3 (25-45 ka), minimal sea ice during the last deglacial (16-11 ka) and early Holocene thermal maximum (11-5 ka) and increasing sea ice during the mid-to-late Holocene (5-0 ka)". Similar interpretations are here presented by Gemery and co-atuhors. The only addition to Cronin et al. (2010) is that "sea-ice cover during the last glacial maximum may have been less extensive at the southern Lomonosov Ridge at our core site (âĽij85.15âŮęN, 152âŮęE) than farther north and towards Greenland", which is pretty weak. Authors' reply: This paper addresses the distribution of key species of benthic Ostracoda and uses them as paleoenvironmental proxies to shed light on benthic community responses to changing ice and ocean conditions during the past 50ka. The core location is in a region of the Arctic unstudied for glacial, deglacial and interglacial paleoceanography and as such, fills an important geographic gap in a region that today is undergoing rapid sea ice decay. Many of the results do corroborate the conclusions of prior studies conducted on other Arctic submarine ridges (Cronin et al., 2010, which focused on a sea ice-dwelling species and Poirier et al. 2012). The new SWERUS core provides evidence for large-scale shifts in ostracode species bathymetric and geographical distributions during rapid climatic transitions. Some evidence suggests that the location of this core may not have been covered by thick ice during the lst glacial period as long as other sites, but we are cautious to state additional studies, especially radiocarbon dating, are needed.

Second, the manuscript is only descriptive and does not present any forcing mechanisms to explain the observed changes in sea ice cover over the past 50,000 years. Why the MIS 3 did not experience perennial sea ice cover when temperatures where

globally lower than during the Late Holocene? What is the link between intermittent perennial and seasonally ice-free conditions during MIS3 and HE/DO? What is the impact of lower sea-level during MIS3 on ocean circulation (less to no North Pacific waters), on sea ice formation (mainly on marginal seas if I am right) and sea ice transport off the Arctic Ocean? The new data should be presented and explained in the context of large scale ocean and atmosphere changes over the past 50,000 years. There are plenty of publications from the GIN Seas and Fram Strait to document NADW inflow (marked here by Krithe spp. and Cytheropteron spp.) and AW outflow (marked here by Polycope spp. and P. caudata). There is also a wealth of publications from continental peri-Arctic to document atmospheric patterns and their impact on central Arctic sea ice. As such, the very attractive title is misleading. Authors' reply: The reviewer poses excellent questions about what atmospheric and oceanic forcings and feedbacks are at play causing sea ice changes. We have added some explanations about the large-scale forcings affecting/controlling/linking oceanographic changes from published literature but a more thorough discussion is beyond the scope of this paper. The regional variability of changes in the sea-ice regime, especially during rapid climatic events, is not yet well understood due to the low sedimentation rates in the central Arctic (1-2 mm/ka).. Our study focused on A. as an indicator for the expansion/contraction of sea ice; other proxies might also be applied to this region (ie, dinoflagellates, IP25).

Third, results are discussed in "climatic phases" that are not congruent with the ostracode faunal changes. It is more sensible to discuss changes in the four "ostracode zones". I however do not fully agree on the four zones. Based on faunal changes more periods can be discussed: The K zone, a first increase in A. arcticum between 42-35 kyrs BPP, a P. caudata peak between 35-27 kyrs BP, a second increase of A. arcticum between 25-20 kyrs BP, a second P. caudata peak between 20-12 kyrs BP, the C zone and the A zone. Line 266-271: The shift between Polycope spp. and the Krithe-Cytheropteron group is at 12 kyrs BP not 14.5 kyrs BP. And the Krithe gp is less than 10%. Is this small increase significant? Over the deglaciation I see the following sequence: P. caudata (20-12 kyrs BP); Cytheropteron (12-9 kyrs BP); Krithe (10-7 kyrs

BP). This is not really discussed. Line 280: Krithe spp. are less than 10%. This is not what I call abundant. Authors' reply: We followed Poirier et al., 2012 faunal zonation, as these zones are well established throughout the Arctic Ocean in 32 cores. Broad deglacial-Holocene faunal changes are discussed an interpreted in Poirier et al. (2012) and further in our paper.

There is no information on why there are so much difference in ostracode abundances and species numbers between the twin cores. Authors' reply: The dominant species' faunal trends in 32MC and 32GC are very similar. The difference in ostracode numbers between the two cores is due to getting the larger sediment sample sizes for the multicore and sampling it every centimeter. For the gravity core, we sampled every 2 or 3 cm but within that interval we sampled usually a smaller amount from half the width from the already halved archive half. This difference is commonplace when splicing together records from two types of cores from the same location. It is preferred as piston and gravity coring often does not recover the uppermost sediments, which in the Arctic can pose a huge problem due to low sed. rates. Hence, one augments the gravity core with a multicore. The sampling strategy is described in the Methods section.

Fourth, the "Results" part present description of results mingled with some environmental interpretations. And the "Discussion" part does not present any environmental interpretations nor forcing mechanisms. The structure should be modified accordingly. Lines 307-325: Useless in the paper. Authors should stick to paleoceanographic reconstructions and interpretations. Authors' reply: We thank the reviewer for pointing this out and we modified the Results section by instead putting all environmental interpretations in the Discussion section. We agree, lines 307-325 are tangential, but are relevant to discussion of microfaunal species indicators of ecosystem regime change.

Fifth, the paper oscillates between presenting new sea ice reconstructions (but no explanation of such changes) and validation of R. mirabilis to infer past sea ice changes. I would say that these are two different topics and should perhaps be presented in two different papers. Authors' reply: R. mirabilis' stratigraphic appearance in intermediate

depth cores is an important finding; they are distinct microfaunal migrational events in which a species hat lives on todays contientnal shelf is found in intervals in sediment cores that is far outside its usual depth and geographic range. For example, CITE ALL CORES This topic is worthy of discussion because it focuses attention on other unique, faunal migrations such as B aculeate AND OTHER. In addition to their paleoceanographic and ecological significance, rapid faunal migrations and limited stratigraphic ranges make these useful stratigraphic marker for correlating cores from across the Arctic Ocean.

Additionally, records of R. mirabilis should be described in the "Results" part. They here appear out of the blue at the very end of paper. Lines 328-330: Ostracode species mentioned here are not presented in the results. There is no way to compare and assess what is written. Although it is difficult to assess here because the records are presented in different plots, it seems to me that R. mirabilis record in the twin cores are similar to the Krithe spp. record with peaks centered at 42-44 kyrs BP and 10-5 ka BP. This contradicts lines 328-333 where authors state that R. mirabilis modern distribution mimics B aculeata's one. This should be expanded. Why these two species share a similar modern distribution (linked to perennial sea ice) while presenting different down-core records whereby B. aculeata is still linked to perennial sea ice while R. mirabilis goes together with species tracking less sea ice and NADW influx into central Arctic? Authors' reply: The reviewer makes excellent points and we have reorganized the Results section and removed comparisons with foraminifera such as B. aculeata.

Sixth, the "Chronology" part is not totally clear to me. Data used to estimate the mentioned 3cm offset between the MC and GC cores are not presented. The tuning below 31.5 cm is not presented. It seems that there is only one point with E. huxleyi to infer the MIS5. I strongly doubt that the mean reservoir age was constant through time. It should be acknowledges even though this may not have a big impact on the results/interpretations here due to low temporal resolution. Authors' reply: We have clarified these points in the text and specified that the reservoir age was not likely constant through time. We used the dominant ostracode patterns to align the MC and GC and thereby determine the 3cm offset. Chronology beyond 50ka and use of E. hux-leyi is presented is based on correlation of sediment properties and dates from other nearby cores. Chronology beyond 50 ka is not relevant to this paper, albeit we still present it as supplementary information for the reader.

Seventh, the "Introduction" is very weak. The scientific issue is not very well presented (only in first and last paragraph). There is not state-of-the art. I suggest to much better highlight the difference to Cronin et al. (2010). Authors' reply: We thank the reviewer for the suggestion to fortify the Introduction. We have added and revised this section accordingly.

---

## Author Comment (AC3) · 3 Jul 2017

A. de Vernal (Referee) devernal.anne@uqam.ca

The manuscript by Gemery et al. addresses an important topic, that of the ocean and climate change in the Arctic during the Quaternary. The new data from the SWERUS

core 32 add useful information on the stratigraphy of ostracods over the last 40,000 years in the Arctic Ocean. The study core is one of the rare relatively well-dated sequence from the central Arctic Ocean, at least for the last 35 kyr and relatively high sedimentation rates (∼1 cm/kyr on average) permit to report the stratigraphical distribution of microfossils with millennial time resolution. The new results from core 32 are very interesting. They are used together with the data from many other cores (most being already published) to present an Arctic Ocean wide synthesis for the last ∼40 kyr. This offers a very valuable contribution as announced in the title and summarized in the abstract. In the manuscript, however, other data encompassing longer time scales, ranging up to the 160 kyr or even 340 kyrs, are discussed with reference to occurrence peaks of Rabimilis mirabilis in the ostracode assemblages. Hence, the scope of the paper is not clear. There is a hiatus between the abstract summarizing the new data from the SWERUS core 32 data and the discussion dealing with the longer time scales. In my opinion, the new data unquestionably deserve publication after a few points is clarified. The comparison with other records encompassing the last 40 kyr is very interesting and could be much useful especially if the basin-scale results are discussed in a more comprehensive manner. The synthesis part on the longer time scales, however, seems to be another story, which would require a better presentation/demonstration of the chronostratigraphy (including uncertainties) before to offer a robust scientific contribution.

My recommendation is therefore to revise the manuscript by focusing on the new data and their implication in term of large-scale paleoceanography at the scale of the last 40 kyrs. The manuscript will then offer an original, robust and useful contribution providing that some clarification/modification are made with regard to (a) the chronology and (b) the absolute abundance of ostracodes. Authors' reply: We thank the reviewer for these suggestions, and to refocus the scope of the paper to the last ∼40ka. The revised copy does this and focuses more on large-scale paleoceanography of the central Arctic Ocean. We reorganized the Results section so that all environmental interpretations are in the Discussion section. Also, in the chronology section we acknowledge that

revised age models may be needed in the future.

(a) The age-depth relationship in cores 32MC and 32G was derived from linear interpolation between 14C dates as shown in figure 2. However, other solutions with highly variable sedimentation rates are very likely in the Arctic Ocean context. In particular, no accumulation or extremely low sedimentation rates during the last glacial maximum are recorded at many sites of the central Arctic Ocean (e.g., Norgaard-Pedersen et al. 2003; Polyak 2004; Not & Hillaire-Marcel 2010; Hanslik et al. 2010). Hence, the age of ca. 20 ka in core 32MC can simply result from mixing. The use of a Bayesian approach (e.g., with the Bacon software for depth/age modelling; Blaauw & Christen, 2011) would be appropriate and could help constraining the uncertainties. Another concern comes for the old 14C ages (> 40 ka) that must be considered with caution because of potential biases due to even extremely small contamination (e.g., Hughen 2007), notably through diagenetic processes and carbonate recrystallisation (Sivan et al., 2002; Douka et al., 2010). Thus, the chronology of the lower part of the sequence, older than about 35 kyr, is equivocal because the absolute age as well as the linear interpolation can be questioned. A critical presentation of the age-depth relationships in the other cores from the Lomonosov and Mendeleev ridges (Figures 4 and 5) would be useful to give an information on the time window represented by the samples analyses, to strengthen the regional zonation proposed and to clearly demonstrate the synchroneity or time lags in the records. Authors' reply: We agree the dates >40ka need to be approached with caution, and we amended the chronology section to note this concern. We agree that a presentation of the age-depth relationships of the other cores from the Lomonosov and Mendeleev ridges (Figures 4 and 5) would be ideal, and we acknowledge in the amended figure caption that the chronology of these cores may NOT have been derived from a standardized method or with the assumptions used to generate 32MC/GC chronology. Our goal of showing these comparison plots of indicator species' abundance was to emphasize the broadly similar faunal changes that occur from core to core in the central Arctic Ocean. Also we specified that the reservoir age was not likely constant through time. Chronology beyond 50ka and use of

E. huxleyi is presented is based on correlation of sediment properties and dates from other nearby cores. Chronology beyond 50 ka is not relevant to this paper, albeit we still present it as supplementary information for the reader.

(b) The results are presented in term of number of ostracod counted and percentages of main taxa. The concentration or density of ostracode valves per unit of weight (g) or unit of volume (cc) would be very useful to describe the real abundance of ostracod in sediment and to get a picture of the actual fluxes of the key species. Moreover, Rabimilis mirabilis is discussed as an important species, but its downcore distribution is not shown in figures 3-5. It should be added (% and concentration) in the diagrams of these figures. Authors' reply: We added the density of ostracodes per gram of sediment to the Appendix, which also lists raw ostracode counts in each sample. The number of ostracode specimens in all 32MC samples exceeded 300, and specimen counts in 32GC ranged from 52 to more than 1000. R. mirabilis' abundance is not plotted in Fig 3 because it is instead presented in Fig.7 along with other cores in which it is found. R. mirabilis' abundance is not plotted in Figs 4 and 5 because it was not present in any of the cores on the Lomonosov Ridge and only in one core on the Mendeleev Ridge (HLY6 in the top 12cm of the core).

Beyond clarification in the presentation of results, some discussion about the actual significance of the ostracodes in the sediment would be helpful, as briefly suggested below. 1. In the interpretative schemes of the result section, the ostracode assemblages are associated with water masses, some of Atlantic origin. Are the ostracodes indicative of actual conditions in bottom waters or to transport with water masses ? Authors' reply: Unlike planktic foraminifers that live in the uppermost water column and are free-floating within water masses, most ostracode species are benthic in habitat and their ecology reflects bottom water environmental conditions. We added a sentence in the introduction to clarify this.

2. Acetabulostoma arcticum is associated with multi-year sea-ice cover, which makes it a very important bio-indicator, actually the only one that can be used to assess "positively" on the occurrence of perennial sea ice as far as I know. The fact that it characterizes the postglacial on the Lomonosov Ridge is important, but its low occurrence during the glacial interval is equivocal. Can it relate to low general productivity due to too thick perennial ice? Its low occurrence on the Mendeleev Ridge for most the study interval is also intriguing. Authors' reply: A. arcticum characterizes perennial sea ice conditions where light can penetrate through the sea ice and surface-ocean productivity is possible. A. arcticum is not benthic dwelling; it lives parasitically in an amphipod that lives under sea ice. We clarified this in Table 4: The stratigraphic distribution of A. arcticum is used as an indicator of periods when the Arctic Ocean experienced thicker sea-ice conditions but not fully glacial conditions when productivity would have halted. This pelagic ostracode is a parasite on Gammarus amphipods that live under sea ice in modern, perennially sea-ice-covered regions in the Arctic (Schornikov, 1970).

3. Rabimilis mirabilis is mentioned as a shallow water taxon. Could it be transported from the shelf (with sea ice for ex.) ? The fact that both adult and juvenial specimens are recovered (lines 361-364) is not a very convincing argument. Authors' reply: The preservation quality of the valves and abundance of valves leads to the hypothesis that spikes in R mirabilis signify abrupt environmental changes. The R. mirabilis valves we found in narrow sediment slices were extremely well preserved. It is highly unlikely that a R. mirabilis population would be transported in such numbers and with excellent preservation as the valves that we found in not just one core but multiple cores on the Lomonosov and Mendeleev Ridges. Transported shells are typically partially dissolved, corroded and/or chalky.

4. The zonation from the Lomonosov Ridge seems relatively robust, but Krithe spp. and Pseudocythere caudata show somewhat different records in the study cores. How can the difference be interpreted ? Does the deeper location of core AOS94-28 matter ? Similar, the assemblages from the Mendeleev Ridge show differences notably with regard to Krithe spp. Pseudocythere caudata. Are the differences indicative of a regionalism? Authors' reply: Usually we find Polycope is inversely correlated with

that of Krithe.Yes, P. caudata's signal could be indicative of regionalism. We include P. caudata because of its fairly robust signal during MIS 3-2 in 32MC/GC. P. caudata appears to be ecologically linked to the surface conditions and generally with A. arcticum and perennial sea ice conditions (per Cronin et al 2014, Fig 6b). The significance of P. caudata could benefit from a DCA or CCA that involves multiple Lomonosov and Mendeleev Ridge cores.

5. High abundance/dominance of Polycope spp. characterizes the pre-Holocene sediment of almost all cores (Figures 3-5). This is interesting as it might indicate uniform water masses from Atlantic origin in intermediate layers of the Arctic Ocean during glacial time. Authors' reply: Yes, Polycope spp. demonstrates a strong signal in most of the cores; it's abundance is inversely correlated with that of Krithe; Polycope becomes dominant ∼ 30-35 ka during late MIS 3 (Cronin et al., 2014, Fig 8). We followed Poirier et al., 2012 faunal zonation, as these zones are well established throughout the Arctic Ocean and in the SWERUS 32 cores. Broad deglacial-Holocene faunal changes are discussed as interpreted in Poirier et al. (2012) and further in our paper.

Other minor comments : - The supplementary tables are not easy to read and there are parts missing. Probably there was a problem when saving them as pdf. Authors' reply: Yes this will be corrected in final production. The spreadsheets were created in excel and exceed the length of a page when converted to a pdf for the review process. - The nomenclature of cores in figures 4 and 5 is not exactly the same than in the map of figure 1, which is a little confusing. Authors' reply: We clarified this in the caption. - In figure 5, the spacing of data points from core HLY6 is so large that comparison with other cores is not very useful ; Linking the data points between _12 ka and _27 ka for core AOS94 8, and between _ 13 ka and 40 ka for core AOS94 12 is inappropriate. Authors' reply: The sampling interval of HLY6 was not as highly resolved compared to the other cores, but is presented because the data are still important in helping to understand the environmental conditions at a millennial scale. We agree about linking data points during a hiatus period or time when a species abundance was zero, so we

have removed the long lines linking the data points from 12 to 27ka and 13 to 40ka.

Please also note the supplement to this comment:
https://www.clim-past-discuss.net/cp-2017-22/cp-2017-22-AC3-supplement.pdf

---

## Author Comment (AC5) · 7 Jul 2017

The paper by Gemery and colleagues represents an interesting study that illustrates how the analysis of ostracod fauna can shed new light on the paleoceanographic

changes occurred in the central Arctic Ocean during the Late Quaternary (ca. the last 50 ka). This study can be particularly welcomed by teams involved in the reconstruction of recent past sea-ice conditions and relative strength of Atlantic Water influx to the Arctic Ocean during periods of climate variations. The Authors also put into question the possibility to use peaks in ostracod species (i.e. Rabilimis mirabilis) commonly recorded at shallow water-depths (<200m) as proxy for abrupt changes in paleoceanographic conditions. In my opinion, the most interesting elements of the paper are (i) the potential paleoceanographic significance of R. mirabilis migration events, (ii) the comparison among relatively high-resolution ostracod data from several cores and (iii) the effort to reconstruct a robust chronological framework for the 2 studied cores (32-GC and 32-MC). However, there are some aspects, concerning the interpretation of ostracod data and text organization, that could be improved:

1. I have the impression that the paper, in its present state, doesn't fully emphasize all the new results derived by the analysis of ostracode fauna (cores 32-GC and 32-MC). In particular, the discussion section (section 5.) only focuses on the distribution of R. mirabilis, while it should also include the reconstruction of paleoceanographic conditions from ca. 50 ka to present (according to the scope and title of the paper), emphasizing the novelty in respect to previous studies undertaken in the same area (e.g.,higher sampling resolution). On the other hand, results concerning R. mirabilis distribution patterns are poorly described in section 4. Moreover, I suggest to describe in more detail the stratigraphic/temporal patterns of ostracod indicator species from the 2 new cores (section 4.2) and more clearly distinguish data interpretation from the discussion and conclusions, based on the comparision among several cores. Authors' reply: We thank the reviewer for this thoughtful and helpful review. Yes, we agree that a restructuring was in order. We find it is more streamlined to present the results of the faunal patterns along with a discussion of their significance so we combined the Results/Discussion section into one and also added a new section (4.6 New faunal events) that presents R. mirabilis migration events and our interpretation.
2. The ostracod zones could be a little bit refined, highlighting the occurrence of a "transitional" ostracod fauna zone, between ca. 42-35 ka, dominated by Polycope spp., but also characterized by remarkable percentages of A. arcticum and Krithe spp.. I think that the ostracod data (Fig. 3) show interesting faunal turnover that could be investigated in depth using a statistical approach. Did the Authors perform multivariate analysis (e.g., DCA) to improve the identification of the main faunal turnover through the core succession/time and the comprehension of the main controlling parameters? Moreover, it could be useful a more detailed explanation of the main turnover in terms of paleoenvironmental conditions: what do the two peaks in P. caudata (between ca. 35-30 ka and 20-12 ka) mean? I also wonder why the percentages of A. arcticum are higher during the mid-late Holocene in respect to the LGM. Authors' reply: We followed Poirier et al., 2012 faunal zonation, as these zones are well established throughout the Arctic Ocean and in the SWERUS 32 cores. Yes, statistical analyses have already been done to establish the ecological relationships of the indicator species with environmental conditions (Gemery et al., 2013; Cronin et al., 1995). We do not interpret P. caudata, but record its frequency. Cronin et al 2014 report that based on P. caudata's co-occurrence with A. arcticum in modern and downcore samples, the benthic species appears to be ecologically linked to the surface conditions (also Cronin et al 2014, Fig 6). Percentages of A. arcticum are higher in the mid-late Holocene than the LGM because sea ice during the glacial at this location may have been too thick to allow light penetration under the ice.

3. The Authors state that the R. mirabilis peaks are composed by in-situ populations because of the presence of well-preserved adult and juvanile valves. I agree with the Authors that this is a good autochthonity indicator, however I wonder if there are other data that can support this interpretation and/or other analyses can be performed to exclude the possibility of resedimentation events. Authors' reply: We feel confident stating that R.mirabilis events represent in-situ populations because of the number and excellent preservation of the specimens. While there is the possibility of resedimentation, we do not see any signs of the shells being reworked.

[Figure]

4. In Mendeleev Ridge area, the visual inspection of ostracod data seems to show a low degree of correlation among cores. Maybe, it could be useful to compare cores ostracod data (Figs. 4, 5) using statistical methods. How much the ostracod patterns are really similar as stated by the Authors (e.g., in the abstract "Comparisons with faunal records from other cores from the Mendeleev and Lomonosov Ridges suggest generally similar patterns, . . .")? Authors' reply: Central Arctic Ocean ostracode faunal patterns documented in publications during the last 20 years (i.e. Cronin et al., 1995) show Cytheropteron spp., Henryhowella asperrima, and Krithe spp. dominate assemblages during the Holocene interglacial period (MIS 1) and interstadial events, while Polycope spp. dominates the glacial period (MIS 2) and stadial events. There are different proportions of these dominant species due to influence of deeper water masses, location, depth of the particular core examined. For example H. asperrima is not found in 32MC/GC but is found in other Lomonsov Ridge cores at deeper depths.

5. The construction of the age-depth model for the 2 new cores deserves a more detailed explaination and discussion. In particular, I'd like to see how ostacod data help to depth align the 2 cores. Authors' reply: To correlate cores 32-MC and 32-GC and produce a composite faunal record, we used patterns in ostracode assemblage in both cores, which led to a 3-cm offset for core 32-GC. It was obvious when comparing the initial depth-abundance plots of 32-MC and 32-GC that the faunal patterns would align if we added 3cm to the GC. After adding the 3-cm offset to sample depths of 32-GC, we applied the 32-MC core chronology down to 31.5 cm core depth (dated at 39.6 ka).

Minor comments: a) Cytheropteron spp. should be added in the abstract along with the other ostracod indicator species. Authors' reply: The Cytheropteron genus includes several deep-water species that are difficult to interpret so we are only including this group in a general way. Generally, the dominance of Krithe and Cytheropteron may signify seasonally open ocean conditions, possibly with deep-water convection as is found in parts of the modern Norwegian-Greenland Seas (Cronin et al., 2013).

b) In the introduction, I suggest to more clearly state the aims of the paper and highlight the novelty of this study in respect to previous works dealing with ostracod fauna from nearby cores. Authors' reply: We added a few sentences in to the introduction referring to previous foundational work.

c) An entire sub-section (5.1.) focused on foraminiferal fauna events is a little bit too much for a paper dealing with ostracodfauna. Authors' reply: We agree, we removed this section.

d) Paleoenvironmental changes documented by ostacodfauna should be reported in conclusions. Authors' reply: We restated the Conclusion section to summarize the general faunal patterns

e) Figure 3: please replace Krithe sp. with Krithe spp. Authors' reply: We only found Krithe hunti in the 32MC and GC cores, but in other cores from the central Arctic, Krithe minima was also found. So for this paper, Krithe sp. refers to Krithe hunti.

Please also note the supplement to this comment: https://www.clim-past-discuss.net/cp-2017-22/cp-2017-22-AC5-supplement.pdf